# SELECTIVE COLLABORATION FOR ROBUST FEDERATED LEARNING

## ABSTRACT

Federated Learning (FL) revolutionizes machine learning by enabling model training across decentralized data sources without aggregating sensitive client data. However, the inherent heterogeneity of client data presents unique challenges, as not all client contributions positively impact model performance. In this work, we propose a novel algorithm, Merit-Based Federated Averaging (`MeritFed`), which dynamically assigns aggregation weights to clients based on their data distribution's relevance to a target objective. By leveraging stochastic gradients and solving an auxiliary optimization problem, our method adaptively identifies beneficial collaborators, ensuring efficient and robust learning. We establish theoretical convergence guarantees under mild assumptions and demonstrate that `MeritFed` achieves superior convergence by harnessing the advantages of diverse yet complementary datasets. Empirical evaluations highlight its ability to mitigate the adverse effects of outlier and adversarial clients, paving the way for more effective and resilient FL in heterogeneous environments.

## 1 INTRODUCTION

Federated Learning (FL) introduces an innovative paradigm redefining traditional machine learning workflow. Instead of centrally pooling sensitive client data, FL allows for model training on decentralized data sources stored directly on client devices (Konečný et al., 2016; Zhang et al., 2021). In this approach, rather than training Machine Learning (ML) models in a centralized manner, a shared model is distributed to all clients. Each client then performs local training, and model updates are exchanged between clients and the FL orchestrator (often referred to as the master server) (McMahan et al., 2017; Shokri & Shmatikov, 2015).

**Personalized Federated Learning (PFL).** The concept of PFL (Collins et al., 2021; Hanzely et al., 2020; Sadiev et al., 2022; Almansoori et al., 2024; Borodich et al., 2021; Sadiev et al., 2022) has been gaining traction. In this framework, each client, often referred to as an agent, takes part in developing their own personalized model variant. This tailored training approach leverages local data distributions, aiming to design models that cater to the distinct attributes of each client's dataset (Fallah et al., 2020). In contrast, standard `Parallel SGD` (Zinkevich et al., 2010) often leads to models that generalize across all clients rather than personalize to the specific data distributions and unique characteristics of individual clients, potentially resulting in suboptimal performance on personalized tasks. However, a prominent challenge arises in this decentralized training landscape due to the data's non-IID (independent and identically distributed) nature across various clients. Data distributions that differ considerably can have a pronounced impact on the convergence and generalization capabilities of the trained models. While certain client-specific data distributions might strengthen model performance, others could prove detrimental, introducing biases or potential adversarial patterns. Additionally, within the personalized federated learning paradigm, the emphasis on crafting individualized models could inadvertently heighten these data disparities (Kairouz et al., 2021). Consequently, this may lead to models that deliver subpar or (potentially) incorrect results when applied to wider or diverse datasets (Kulkarni et al., 2020).

**Collaboration as a service.** In this paper, we introduce a modified protocol for FL that deviates from a strictly personalized approach. Rather than focusing solely on refining individualized models, our approach seeks to harness the advantages of distinct data distributions, curb the detrimental effects of outlier clients, and promote collaborative learning. Through this innovative training mechanism,

our algorithm discerns which clients are optimal collaborators to ensure faster convergence and potentially better generalization.

## 1.1 SETUP

We assume that there are $n$ clients participating in the training and consider the first one as a target client. The goal is to train the model for this client, i.e., we consider

$$\min_{x \in \mathbb{R}^d} \left\{ f(x) \equiv f_1(x) := \mathbb{E}_{\xi_1 \sim \mathcal{D}_1}[f_{\xi_1}(x)] \right\}, \tag{1}$$

where $f_{\xi_1} : \mathbb{R}^d \to \mathbb{R}$ is the loss function on sample $\xi_1$ and $f : \mathbb{R}^d \to \mathbb{R}$ is an expected loss. Other clients can also have data sampled from similar distributions, but we also allow adversarial participants, e.g., Byzantines (Lamport et al., 1982; Lyu et al., 2020). That is, some clients can be beneficial for the training in certain stages, but they are not assumed to be known apriori.

The considered target client scenario naturally arises in *cross-silo* FL on medical image data. In such applications, different hospitals naturally have different data distributions (e.g., due to the differences in the equipment). Therefore, the data coming from one clinic can be useless to another clinic. At the same time, several clinics can have similar data distributions.

While our setup focuses on optimizing models for a single target client, an alternative direction involves using a small IID validation set at the server to guide the aggregation of updates from locally Non-IID clients. This can help produce a model that performs well across clients, as if the data were IID. This formulation may be more broadly applicable in practice when personalization is not the goal and such a validation set is available. Our framework can naturally extend to this setting by changing the objective function. In contrast, our setup is more privacy-aware: clients share neither their raw data with the server or other clients nor pure stochastic gradients, making our method particularly suited for sensitive applications.

## 1.2 CONTRIBUTION

Our main contributions are listed below.

- **New method: `MeritFed`.** We proposed a new method called Merit-based Federated Averaging for Diverse Datasets (`MeritFed`) that aims to solve (1). The key idea is to use the stochastic gradients received from the clients to adjust the weights of averaging through the inexact solving of the auxiliary problem of minimizing a validation loss as a function of aggregation weights.
- **Provable convergence under mild assumptions.** We prove that `MeritFed` converges not worse than `SGD` that averages only the stochastic gradients (or pseudo-gradients for multiple local steps) received from clients having the same data distribution (these clients are not known apriori) for smooth non-convex and Polyak-Lojasiewicz functions under standard bounded variance assumption (Theorem 1). We also prove that `MeritFed` has even better theoretical convergence when there exists a group of clients with "close enough" data distribution (Theorem 2).
- **Utilizing all possible benefits.** We numerically show that `MeritFed` benefits from collaboration with clients having different yet close to the target one data distributions. That is, `MeritFed` automatically detects beneficial clients at any stage of training. Moreover, we illustrate the Byzantine robustness of the proposed method even when Byzantine workers form a majority.

## 1.3 RELATED WORK

**Federated optimization.** Standard results in distributed/federated optimization focus on the problem:

$$\min_{x \in \mathbb{R}^d} \frac{1}{n} \sum_{i=1}^{n} f_i(x), \tag{2}$$

where $f_i(x)$ represents either expected or empirical loss on the client $i$. This problem significantly differs from (1), since one cannot completely ignore the updates from some clients to achieve a better solution. Typically, in this case, communication is the main bottleneck of the methods for solving such problems. To address this issue one can use communication compression (Alistarh et al., 2017; Stich et al., 2018; Mishchenko et al., 2019), local steps (Stich, 2018; Khaled et al., 2020; Kairouz et al., 2021; Wang et al., 2021; Mishchenko et al., 2022; Sadiev et al., 2022; Beznosikov et al., 2024), client importance sampling (Cho et al., 2020; Nguyen et al., 2020; Ribero & Vikalo, 2020; Lai et al., 2021; Luo et al., 2022; Chen et al., 2022d), or decentralized protocols (Lian et al., 2017; Song et al.,

2022), or FL of graph neural network on graph data (Tan et al., 2023). However, these techniques are orthogonal to what we focus on in our paper, though incorporating them into our algorithm is a prominent direction for future research.

**Clustered FL.** Another way of utilizing benefits from the other clients is the clustering of clients based on some information about their data or personalized models. (Tang et al., 2021) propose a personalized formulation with $\ell_2$-regularization that attracts a personalized model of a worker to the center of the cluster that this worker belongs to. A similar objective is studied by (Ma et al., 2022). (Ghosh et al., 2020) develop an algorithm that updates clusters's centers using the gradients of those clients that have the smallest loss functions at the considered cluster's center. It is worth mentioning that, in contrast to our work, the mentioned works modify the personalized objective to illustrate some benefits of collaboration while we focus on the pure personalized problem of the target client. Under the assumption that the data distributions of each client are mixtures of some finite set of underlying distributions, (Marfoq et al., 2021) derive the convergence result for the Federated Expectation-Maximization algorithm. This is the closest work to our setup in the Clustered FL literature. However, in contrast to (Marfoq et al., 2021), we do not assume that the gradients are bounded and that the local loss functions have bounded gradient dissimilarity. Another close work to ours is (Fraboni et al., 2021), where the authors consider so-called clustered-based sampling. However, (Fraboni et al., 2021) also make a non-standard assumption on the bounded dissimilarity of the local loss functions, while one of the key properties of our approach is its robustness to arbitrary clients' heterogeneity. (Li et al., 2020) is also a relevant paper in the sense that not all workers are selected for aggregation at each communication round (due to the client sampling). However, this work focuses on weighted empirical risk minimization (with weights proportional to the dataset size), i.e., (Li et al., 2020) consider a different problem. Ma et al. (2023) addresses the "clustering collapse" issue with clustering rules based on the min-loss criterion and k-means style criterion.(Bao et al., 2023) focus on optimizing collaboration in federated learning by grouping workers into clusters based on data similarity. Their method requires minimizing a score function for each pair of clients to measure the distance between their data. This clustering process involves computational efforts during the preprocessing stage, and the training within each cluster uses static aggregation weights.

**Non-uniform averaging.** There are works studying the convergence of distributed `SGD`-type methods that use non-uniform, fixed weights of averaging. (Ding & Wang, 2022) propose a method to detect collaboration partners and adaptively learn "several" models for numerous heterogeneous clients. Directed graph edge weights are used to calculate group partitioning. Since the calculation of optimal weights in their approach is based on similarity measures between clients' data, it is unclear how to compute them in practice without sacrificing the privacy. (Even et al., 2022) develop and analyze another approach for personalized aggregation, where each client filters gradients and aggregates them using fixed weights. The optimal weights also require estimating the distance between distributions (or communicating empirical means among all clients and estimating effective dimensions). Both works do not consider weights evolving in time, which is one of the key features of our method.

Non-fixed weights are considered in (Wu & Wang, 2021), but the authors focus on non-personalized problem formulation. In particular, (Wu & Wang, 2021) propose the method called `FedAdp` that uses cosine similarity between gradients and the Gompertz function for updating aggregation weights. Under the strong bounded local gradient dissimilarity assumption[1], (Wu & Wang, 2021) derive a non-conventional upper bound (for the loss function at the last iterate of their algorithm) that does not necessarily imply convergence of the method. (Zhang et al., 2020) introduce `FedFomo` that uses additional data to adjust the weights of aggregation in Federated Averaging. In this context, `FedFomo` is close to `MeritFed`. However, the weights selection formulas significantly differ from ours. In particular, (Zhang et al., 2020) do not relate the proposed weights with the minimization problem from Line 9 of our method. In addition, there is no theoretical convergence analysis of `FedFomo`.

**Bi-level optimization.** Taking into account that we want to solve problem (1) using the information coming from not only the target client, it is natural to consider the following bi-level optimization

---

[1](Wu & Wang, 2021) assume that there exist constants $A, B > 0$ such that $A\|\nabla f(x)\| \leq \|\nabla f_i(x)\| \leq B\|\nabla f(x)\|$ for every client $i \in [n]$ and any $x$, where $f(x) = \frac{1}{n}\sum_{i=1}^{n} f_i(x)$.

(BLO) problem formulation:

$$\min_{w \in \Delta_1^n} \quad f(x^*(w)), \tag{3}$$

$$\text{s.t.} \quad x^*(w) \in \arg\min_{x \in \mathbb{R}^d} \sum_{i=1}^n w_i f_i(x), \tag{4}$$

where $\Delta_1^n$ is a unit simplex in $\mathbb{R}^n$: $\Delta_1^n = \{w \in \mathbb{R}^n \mid \sum_{i=1}^n w_i = 1, \ w_i \geq 0 \ \forall i \in [n]\}$. The problem in (3) is usually called the upper-level problem (UL), while the problem in (4) is the lower-level (LL) one. Since in our case $f(x) \equiv f_1(x)$, (3)-(4) is equivalent to (1). In the general case, this equivalence does not always hold and, in addition, function $f$ is allowed to depend on $w$ not only through $x^*$. All these factors make the general BLO problem hard to solve. The literature for this general class of problems is quite rich, and we cover only closely related works.

The closest works to ours are (Chen et al., 2021a), which propose so-called Target-Aware Weighted Training (TAWT), and its extension to the federated setup (Huang et al., 2022). Their analysis relies on the existence of weights $w$, such that $\text{dist}(\sum_{i=1}^n w_i \mathcal{D}_i, \mathcal{D}_{\text{target}}) = 0$ in terms so-called representation-based distance (Chen et al., 2021a), which is also zero in our case, or existence of identical neighbors. However, the analysis is based on BLO's techniques and requires a hypergradient estimation, i.e., $\nabla_w f(x^*(w), w)$, which is usually hard to compute. To avoid the hypergradient calculation, (Chen et al., 2021a) also propose a heuristic based on the usage of cosine similarity between the clients' gradients, which makes the implementation of the algorithm similar to FedAdp (Wu & Wang, 2021).

In fact, there are two major difficulties in estimating hypergradient. The first one is that the optimal solution $x^*(w)$ of the lower problem for every given $w$ needs to be estimated. The known approaches iteratively update the lower variable $x$ multiple times before updating $w$, which causes high communication costs in a distributed setup. A lot of methods (Ghadimi & Wang, 2018; Hong et al., 2020; Chen et al., 2021b; Ji et al., 2021; 2022) are proposed to effectively estimate $x^*(w)$ before updating $w$, but anyway the less precise estimate slows down the convergence. The second obstacle is that hypergradient calculation requires second-order derivatives of $f_i(w, x)$. Many existing methods (Chen et al., 2022c; Dagréou et al., 2022) use an explicit second-order derivation of $f_i(w, x)$ with a major focus on efficiently estimating its Jacobian and inverse Hessian, which is computationally expensive itself, but also dramatically increases the communication cost in a distributed setup. A number of methods (Chen et al., 2022c; Li et al., 2022; Dagréou et al., 2022) avoid directly estimating its second-order computation and only use the first-order information of both upper and lower objectives, but they still have high communication costs and do not exploit our assumptions. For a more detailed review of BLO, we refer to (Zhang et al., 2023; Liu et al., 2021; Chen et al., 2022a).

## 2 MERITFED: MERIT-BASED FEDERATED LEARNING FOR DIVERSE DATASETS

Recall that the primary objective the target client seeks to solve is given by (1) where $n$ workers are connected with a parameter-server. Standard Parallel SGD

$$x^{t+1} = x^t - \frac{\gamma}{n} \sum_{i=1}^n g_i(x^t, \boldsymbol{\xi}_i), \tag{5}$$

where $g_i(x^t, \boldsymbol{\xi}_i)$ denotes a stochastic gradient (unbiased estimate of $\nabla f_i(x^t)$) received from client $i$, cannot solve problem (1) in general, since workers $\{2, \ldots, n\}$ do not necessarily have the same data distribution as the target client. This issue can be solved if we modify the method as follows:

$$x^{t+1} = x^t - \frac{\gamma}{|\mathcal{G}|} \sum_{i \in \mathcal{G}} g_i(x^t, \boldsymbol{\xi}_i), \tag{6}$$

where $\mathcal{G}$ denotes the set of workers that have the same data distribution as the target worker. However, the group $\mathcal{G}$ is not known in advance. This aspect makes the method from (6) impractical. Moreover, this method ignores potentially useful vectors received from the workers having different yet similar data distributions.

### 2.1 THE PROPOSED METHOD

We develop Merit-based Federated Learning for Diverse Datasets (MeritFed; see Algorithm 1) aimed at solving (1) and safely gathering all potential benefits from collaboration with other clients. As in Parallel SGD all clients are required to send stochastic gradients to the server. However, in

---

**Algorithm 1** `MeritFed`: Merit-based Federated Learning for Diverse Datasets

---

1: **Input:** Starting point $x^0 \in \mathbb{R}^d$, stepsize $\gamma > 0$
2: **for** $t = 0, \dots$ **do**
3:     server sends $x_{i,0}^t = x^t$ to each worker
4:     **for** each worker $i = 1, \dots, n$ **in parallel do**
5:         **for** $k = 0, \cdots, K-1$ **do**                $\boxed{\text{If } K = 1}$
6:             compute stoch. gradient $g_{i,k}(x_{i,k}^t, \boldsymbol{\xi}_{i,k})$ from local data
7:              $x_{i,k+1}^t = x_{i,k}^t - \gamma_l g_{i,k}^t$            $\boxed{\gamma_l = 1}$
8:         **send** $\Delta_i^t = x_{i,K}^t - x^t$ to the server     $\boxed{\Delta_i^t = -g_i(x^t, \boldsymbol{\xi}_i)}$

9:    $w^{t+1} \approx \underset{w \in \Delta_1^n}{\arg\min} f\left(x^t + \gamma \sum\limits_{i=1}^n w_i \Delta_i^t\right)$    $\boxed{w^{t+1} \approx \underset{w \in \Delta_1^n}{\arg\min} f\left(x^t - \gamma \sum\limits_{i=1}^n w_i g_i(x^t, \boldsymbol{\xi}_i)\right)}$

10:   $x^{t+1} = x^t + \gamma \sum\limits_{i=1}^n w_i^{t+1} \Delta_i^t$              $\boxed{x^{t+1} = x^t - \gamma \sum\limits_{i=1}^n w_i^{t+1} g_i(x^t, \boldsymbol{\xi}_i)}$

---

contrast to uniform averaging of the received stochastic gradients, `MeritFed` uses the weights $w^t$ from the unit simplex $\Delta_1^n$ that are updated at each iteration. In particular, the new vector of weights $w^{t+1} \in \mathbb{R}^n$ at iteration $t$ approximates $\arg\min_{w \in \Delta_1^n} f(x^t - \gamma \sum_{i=1}^n w_i g_i(x^t, \boldsymbol{\xi}_i))$, where $K = 1$ for simplicity. Then, the server uses the weights for averaging stochastic gradients and updating $x^t$.

**Local steps.** Our approach provably supports multiple local updates. The results are given by Theorem 1. But some results and experiments are presented for $K = 1$ for the sake of simplicity.

## 2.2 AUXILIARY PROBLEM IN LINE 9

In general, solving the problem in Line 9 is not easier than solving the original problem (1). Instead, MeritFed requires solving it *approximately*. That is, the dataset used for solving this problem only needs to have the same distribution as the target client's data. In particular, if the training data of the target client is sufficiently good to approximate the expected loss function $f$, then no extra data is required. Theoretically, the validation data only needs to have the same distribution as the target client's data, so validation data can be the same as the training data (or duplicate them). Sections 4 and D shows experimental results where the validation data duplicates the training data. Moreover, the validation dataset size is much smaller than the training dataset in our experiments. Alternative approach dividing the training data into two sets is described in Section A.4.

To avoid any risk of compromising clients' privacy, the target client dataset should be stored only on the target client, and stochastic gradients received from other clients cannot be directly sent to the target client. To satisfy these requirements, one can approximate

$$\arg\min_{w \in \Delta_1^n} \left\{\varphi_t(w) \equiv f\left(x^t - \gamma \sum\nolimits_{i=1}^n w_i g_i(x^t, \boldsymbol{\xi}_i)\right)\right\} \tag{7}$$

using *zeroth-order*[2] Mirror Descent (or its accelerated version) (Duchi et al., 2015; Shamir, 2017; Gasnikov et al., 2022b):

$$w^{k+1} = \arg\min_{w \in \Delta_1^n} \left\{\alpha \langle \tilde{g}^k, w\rangle + D_r(w, w^k)\right\}, \tag{8}$$

where $\alpha > 0$ is the stepsize, $\tilde{g}^k$ is a finite-difference approximation of the directional derivative of sampled function

$$\varphi_{t,\xi^k}(w) \overset{\text{def}}{=} f_{\xi^k}\left(x^t - \gamma \sum\nolimits_{i=1}^n w_i g_i(x^t, \boldsymbol{\xi}_i)\right), \tag{9}$$

where $\xi^k$ is a fresh sample from the distribution $\mathcal{D}_1$ independent from all previous steps of the method, e.g., one can use $\tilde{g}^k = \frac{n(\varphi_{t,\xi^k}(w^k + he) - \varphi_{t,\xi^k}(w^k - he))}{2h}$ for $h > 0$ and $e$ being sampled from the uniform distribution on the unit Euclidean sphere, and $D_r(w, w^k) = r(w) - r(w^k) - \langle \nabla r(w^k), w - w^k\rangle$ is the Bregman divergence associated with a 1-strongly convex function $r$. Although, typically,

---

[2]In this case, the server can ask the target client to evaluate loss values at the required points without sending the stochastic gradients received from other workers.

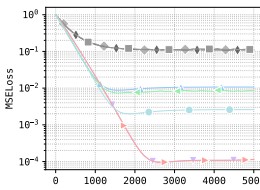 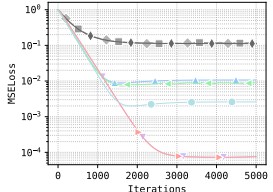 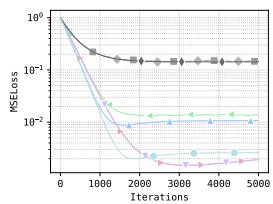 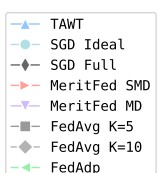

Figure 1: Mean Estimation: $\mu = 0.001$, MD learning rate = 3.5.

Figure 2: Mean Estimation: $\mu = 0.01$, MD learning rate = 4.5.

Figure 3: Mean Estimation: $\mu = 0.1$, MD learning rate = 12.5.

the oracle complexity bounds for gradient-free methods have $\mathcal{O}(n)$ dependence on the problem dimension (Gasnikov et al., 2022a), one can get just $\mathcal{O}(\log^2(n))$, in the case of the optimization over the probability simplex (Shamir, 2017; Gasnikov et al., 2022b). More precisely, if $f$ is $M_2$-Lipschitz w.r.t. $\ell_2$-norm and convex, then one can achieve $\mathbb{E}[\varphi_t(w) - \varphi_t(w^*)] \leq \delta$ using $\mathcal{O}\big(M_2^2 \log^2(n)/\delta^2\big)$ computations of $\varphi$, where $R$ is $\ell_1$-distance between the starting point and the solution (Gasnikov et al., 2022b) and prox-function $r(w) = \sum_{i=1}^n w_i \log(w_i)$, which is 1-strongly convex w.r.t. $\ell_1$-norm.

**Memory usage.** It is worth mentioning that `MeritFed` requires the server to store $n$ vectors at each iteration for solving the problem in Line 9. While standard `SGD` does not require such a memory, closely related methods — `FedAdp` and `TAWT` — also require the server to store $n$ vectors for the computation of the weights for aggregation. However, for modern servers, this is not an issue.

## 3 CONVERGENCE ANALYSIS

In our analysis, we rely on the standard assumptions for non-convex optimization literature.

**Assumption 1.** *For all $i \in \mathcal{G}$ the stochastic gradient $g_i(x, \boldsymbol{\xi}_i)$ is an unbiased estimator of $\nabla f_i(x)$ with bounded variance, i.e., $\mathbb{E}_{\boldsymbol{\xi}_i}[g_i(x, \boldsymbol{\xi}_i)] = \nabla f_i(x)$ and for $\sigma_{\mathcal{G}} \geq 0$*

$$\mathbb{E}_{\boldsymbol{\xi}_i}\|g_i(x, \boldsymbol{\xi}_i) - \nabla f_i(x)\|^2 \leq \sigma_{\mathcal{G}}^2. \tag{10}$$

*Moreover, $f$ is $L$-smooth, i.e., $\forall\, x, y \in \mathbb{R}^d$*

$$f(x) \leq f(y) + \langle \nabla f(y), x - y \rangle + \frac{L}{2}\|x - y\|^2. \tag{Lip}$$

The above assumption combines the well-known bounded variance and smoothness of the objective assumptions. It is classical for the analysis of stochastic optimization methods, e.g., see (Nemirovski et al., 2009; Juditsky et al., 2011).

Next, we assume that there exists a set of workers with "close enough" data distributions to the target one. This can be formalized as follows.

**Assumption 2.** *Let $\mathcal{F} \subseteq [n]$ be a subset of workers such that $\mathcal{F} \cap \mathcal{G} = \varnothing$ and for some $\nu \geq 0$, $\rho \geq 0$ and all $x \in \mathbb{R}^d$*

$$\left\| \frac{1}{F} \sum_{i \in \mathcal{F}} \nabla f_i(x) - \nabla f(x) \right\|^2 \leq \nu \|\nabla f(x)\|^2 + \rho^2. \tag{11}$$

*Moreover, for all $i \in \mathcal{F}$, the stochastic gradient $g_i(x, \boldsymbol{\xi}_i)$ is an unbiased estimator of $\nabla f_i(x)$ with bounded variance, i.e., $\mathbb{E}_{\boldsymbol{\xi}_i}[g_i(x, \boldsymbol{\xi}_i)] = \nabla f_i(x)$ and for $\sigma_{\mathcal{F}} \geq 0$*

$$\mathbb{E}_{\boldsymbol{\xi}_i}\|g_i(x, \boldsymbol{\xi}_i) - \nabla f_i(x)\|^2 \leq \sigma_{\mathcal{F}}^2.$$

The above assumption guarantees that the gradients from workers in $\mathcal{F}$ approximate the true global gradient within relative and absolute error bounds and the stochastic gradients from these workers also have bounded variance. In practice, $\nu$ and $\rho$ can depend on $x$, and can be relatively small if $x$ is far from the solution. However, for simplicity of the analysis we assume that $\nu$ and $\rho$ are constants.

Finally, we also make the following (optional) assumption called Polyak-Łojasiewicz (PŁ) condition (Polyak, 1963; Lojasiewicz, 1963).

**Assumption 3.** *$f$ satisfies Polyak-Łojasiewicz (PŁ) condition with parameter $\mu$, i.e., for $\mu \geq 0$*

$$f^* \geq f(x) - \frac{1}{2\mu}\|\nabla f(x)\|^2, \quad \forall\, x \in \mathbb{R}^d. \tag{PL}$$

This assumption belongs to the class of structured non-convexity conditions allowing linear convergence for first-order methods, e.g., Gradient Descent (Necoara et al., 2019).

The main result for `MeritFed` is given below (see the proof in Appendix B).

**Theorem 1.** *Let Assumptions 1 holds. Then after $T$ iterations, if $K = 1$ `MeritFed` with $\gamma \leq \frac{1}{2L}$ outputs $x^t$, $t = 0, \cdots, T-1$ such that*

$$\frac{1}{T} \sum_{t=0}^{T-1} \mathbb{E}\|\nabla f(x^t)\|^2 \leq \frac{2\left(f\left(x^0\right) - f(x^*)\right)}{T\gamma} + \frac{2\sigma^2 \gamma L}{G} + \frac{2\delta}{\gamma},$$

*and if $K > 1$ `MeritFed` with $\gamma = 2$, $\gamma_l \leq \frac{1}{12LK}$ outputs $x^t$, $t = 0, \cdots, T-1$*

$$\frac{1}{T} \sum_{t=0}^{T-1} \mathbb{E}\|\nabla f(x^t)\|^2 \leq \frac{4\left(f(x^0) - \mathbb{E}f(x^T)\right)}{\gamma_l K T} + 24\gamma_l^2 K L^2 \sigma_G^2 + \frac{32\gamma_l L \sigma_G^2}{G} + \frac{4\delta}{\gamma_l K},$$

*where $\delta$ is the accuracy of solving the problem in Line 9 and $G = |\mathcal{G}|$. Moreover if Assumption 3 additionally holds, if $K = 1$ `MeritFed` with $\gamma \leq \frac{1}{2L}$ outputs $x^T$ such that*

$$\mathbb{E}[f\left(x^T\right) - f^*] \leq (1 - \gamma\mu)^T \left(f\left(x^0\right) - f^*\right) + \frac{\sigma^2 \gamma L}{\mu G} + \frac{\delta}{\gamma\mu},$$

*and if $K > 1$ `MeritFed` with $\gamma = 2$, $\gamma_l \leq \frac{1}{12LK}$ outputs $x^t$, $t = 0, \cdots, T-1$*

$$\mathbb{E}[f(x^T) - f^*] \leq \left(1 - \frac{\mu\gamma_l K}{2}\right)^T [f(x^0) - f^*] + \frac{12\gamma_l^2 K L^2 \sigma_G^2}{\mu} + \frac{16\gamma_l L \sigma_G^2}{\mu G} + \frac{2\delta}{\mu\gamma_l K}.$$

If $\delta$ is small, then the above result matches the known results for Parallel `SGD` (Ghadimi & Lan, 2013; Karimi et al., 2016; Khaled & Richtárik, 2022) that uniformly averages the workers from the group $\mathcal{G}$, i.e., those workers that have data distribution $\mathcal{D}_1$ (see the method in (6)). In fact, we see a linear speed-up of $1/G$ in the obtained convergence rates.

Note, that when $K > 1$ the terms with no linear speed-up contain $\gamma_l$ with a higher power, that recovers results for `Local SGD` and implies that the terms vanish faster with vanishing stepsize.

Moreover, in the case when some workers have different yet similar data, which we formalize as Assumption 2, we provide an improved result below. We consider $K = 1$ for the sake of simplicity.

**Theorem 2.** *Let Assumptions 1 and 2 hold with $G = |\mathcal{G}|$, $F = |\mathcal{F}|$, $\nu \leq \frac{G}{F}$. Then after $T$ iterations of `MeritFed` with $\gamma \leq \frac{1}{8L}$ outputs $x^t$, $t = 0, \cdots, T-1$ such that*

$$\frac{1}{T} \sum_{t=0}^{T-1} \mathbb{E}\|\nabla f(x^t)\|^2 \leq \frac{4\left(f\left(x^0\right) - f(x^*)\right)}{T\gamma} + 2\min\left(\frac{\sigma_\mathcal{G}^2 \gamma L}{G} + \frac{\delta}{\gamma}, \frac{4\gamma LG\sigma_\mathcal{G}^2}{(G+F)^2} + \frac{4\gamma LF\sigma_\mathcal{F}^2}{(G+F)^2} + \frac{\rho^2 F}{G+F} + \frac{2\delta}{\gamma}\right),$$

*where $\delta$ is the accuracy of solving the problem in Line 9. Moreover if Assumption 3 additionally holds, then after $T$ iterations of `MeritFed` with $\gamma \leq \frac{1}{8L}$ outputs $x^T$ such that*

$$\mathbb{E}f\left(x^T\right) - f^* \leq (1 - \gamma\mu)^T \left(f\left(x^0\right) - f^*\right) + \frac{1}{\mu}\min\left(\frac{\sigma_\mathcal{G}^2 \gamma L}{G} + \frac{\delta}{\gamma}, \frac{4\gamma LG\sigma_\mathcal{G}^2}{(G+F)^2} + \frac{4\gamma LF\sigma_\mathcal{F}^2}{(G+F)^2} + \frac{\rho^2 F}{G+F} + \frac{2\delta}{\gamma}\right).$$

Assumption 2 is reasonable, especially at the initial stage of training when the generated points are far from the solution (see the discussion after Assumption 2). So the theorem shows that the variance-induced term is reduced, allowing for a linear speedup proportional to $1/(G+F)$, compared to $1/G$ without the assumption (Theorem 1). Moreover, if $\rho$ and $\delta$ are small, then the neighborhood term $\mathcal{E}$ is smaller than the neighborhood term from Theorem 1 and, consequently, than the neighborhood term in the convergence bound for the method from (6). Theorem 2 also implies that for small $\delta$ `MeritFed` converges not worse than Parallel `SGD` that uniformly averages the workers from $\mathcal{G} \cup \mathcal{F}$.

`MeritFed` needs neither identifying distribution-similar workers nor high-precision solving of Line 9, and empirically converges well even when workers' distributions are distinct but close.

## 4 EXPERIMENTS

Since the literature on FL is very rich, we focus only on the closely related methods satisfying two criteria: (i) they solve the same problem as we consider in 1, and (ii) have theoretical guarantees. That is, we evaluate the performance of proposed methods in comparison with `FedAdp` (Wu &

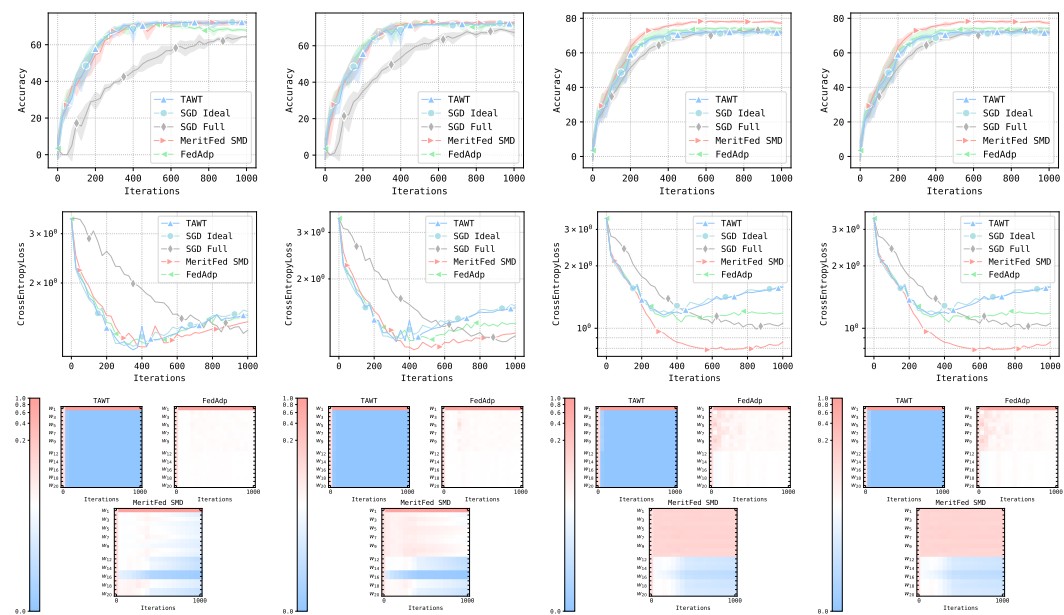

Figure 4: GoEmotions (extra val.): $\alpha = 0.5$
Figure 5: GoEmotions (extra val.): $\alpha = 0.7$
Figure 6: GoEmotions (extra val.): $\alpha = 0.9$
Figure 7: GoEmotions (extra val.): $\alpha = 0.99$

Wang, 2021), `TAWT` (Chen et al., 2021a), and `FedProx` (Li et al., 2020) (`FedProx` reduces to `FedAvg` if there are no local steps, that is the setup for `MeritFed`). We also compare standard SGD with uniform weights (labeled as `SGD Full`[3]), `SGD` that collects only gradients from clients with the target distribution (`SGD Ideal`) and two versions of our algorithm: (i) `MeritFed SMD`, samples gradient for the Mirror Descent subroutine, and (ii) `MeritFed MD`, that uses the full dataset (additional or train) to calculate gradient for Mirror Descent step. In addition, we present the evolution of weights (if applicable) using heat-map plots. In the main text, we show the results for the case when the additional validation dataset is available for the problem in Line 9. Additional experiments and details with the usage of train data for the problem in Line 9, with the presence of Byzantine participants and with more workers, are provided in Appendix D.

**Mean estimation.** The problem is to find such a vector that minimizes the mean squared distance to the data samples. More formally, the goal is to solve $\min_{x \in \mathbb{R}^d} \mathbb{E}_{\xi \sim \mathcal{D}_1} \|x - \xi\|^2$, that has the optimum at $x^* = \mathbb{E}_{\xi \sim \mathcal{D}_1}[\xi]$. We consider $\mathcal{D}_1 = \mathcal{N}(0, \boldsymbol{I})$ and also two other distributions from where some clients also get samples: $\mathcal{D}_2 = \mathcal{N}(\mu \boldsymbol{1}, \boldsymbol{I})$ and $\mathcal{D}_3 = \mathcal{N}(e, \boldsymbol{I})$, where $\boldsymbol{1} = (1, 1, \ldots, 1)^\top \in \mathbb{R}^d$, $\mu > 0$ is a parameter, and $e$ is some vector that we obtain in advance via sampling uniformly at random from the unit Euclidean sphere. Detailed experimental setup is provided in Section D.2.

We consider three cases: $\mu = 0.001, 0.01, 0.1$. The smaller $\mu$ is, the closer $\mathcal{D}_2$ is to $\mathcal{D}_1$ and, thus, the more beneficial the samples from the second group are. Therefore, for small $\mu$, we expect to see that `MeritFed` outperforms `SGD Ideal`. Moreover, since the workers from the third group have quite different data distribution, `SGD Full` is expected to work worse than other baselines.

The results are presented in Figures 1-3. They fit the described intuition and our theory well: the workers from the second group are beneficial (since their distributions are close enough to the distribution of the target client). Indeed, `MeritFed` achieves better optimization error (due to the smaller variance because of the averaging with more workers). However, when the dissimilarity between distributions is large the second group becomes less useful for the training, and `MeritFed` has comparable performance to `SGD Ideal` and consistently outperforms other methods.

**Texts classification: GoEmotions + BERT.** The next problem we consider is devoted to fine-tuning pretrained BERT (Devlin et al., 2018) model for emotions classification on the GoEmotions dataset (Demszky et al., 2020). The dataset consist of texts labeled with one or more of 28 emotions. First of all, we form "truncated dataset" by cutting the dataset so that its each entry has the only label.

---

[3]Although, `FedProx` and `SGD Full` are designed for standard empirical risk minimization; these are our standard baselines.

Then we use Ekman mapping (Ekman, 1992) to split the data between clients. According to the mapping, 28 emotions can be mapped to 7 basic emotions. That is, we simulate a situation when the target client classifies only basic emotions, e.g., the target client has only emotions belonging to "joy" class and namely includes only "joy", "amusement", "approval", "excitement", "gratitude", "love", "optimism", "relief", "pride", "admiration", "desire", "caring". The distribution of these sub-emotions is kept to be the same as the distribution of the truncated train dataset. Clients, that data are supposed to have similar distribution (second group – next 10 clients), also have texts from base class "joy" and are labeled as one of the sub-emotion belonging to "joy". The distribution of sub-emotions is also the same as the distribution of the truncated train dataset. These texts constitute an $\alpha$ portion of the total client's data. The other $1 - \alpha$ portion of the texts is taken from "neutral" class. The rest of clients (third group – next 9 clients) are supposed to have different distribution and their data consist of either texts belonging to one of the other basic emotion, either mixed with neutral (if there is not enough texts to have a desired number of samples) or texts from "neutral" class only. Again, the distribution of sub-emotions is the same as the distribution of the truncated train dataset. The results are presented in Figures 4-7. The target client benefits from collaborating with clients from the second group and achieves better accuracy using `MeritFed`. See Section D.3 for the detailed description.

**MedMNIST.** We apply `MeritFed` to enhance the classification of medical images, as introduced in the MedMNIST dataset (Yang et al., 2021). MedMNIST offers medical image datasets, including three datasets featuring images of internal organs (Organ{A,C,S}MNIST) with identical labels. These datasets can be collectively utilized during training to improve accuracy. A potential method involves aggregating gradients computed from these three datasets. However, due to the diverse nature of the data, some datasets may have limited contributions to the training. We anticipate that adaptive aggregation, provided by `MeritFed`, will improve the model's performance. For empirical justification, we assume that each worker possesses one MedMNIST dataset. Importantly, `MeritFed` does not restrict the setup to only three workers and accommodates additional clients with irrelevant data, aligning with real-world scenarios. To demonstrate this, we introduce a nuisance worker handling data from other MedMNIST datasets. See Appendix D.4 for the detailed description.

**Image classification: CIFAR10 + ResNet18.** The results can be found in Appendix D.6.

## 5 CONCLUSION

We introduced a novel algorithm called Merit-based Federated Learning (`MeritFed`) to address the challenges posed by the heterogeneous data in federated learning via the adaptive selection of the aggregation weights through solving the auxiliary problem at each iteration. We showed that `MeritFed` can effectively harness the advantages of distinct data distributions, control the detrimental effects of outlier clients, and promote collaborative learning. We assign adaptive aggregation weights to clients participating in training, allowing for faster convergence and potentially better generalization. `MeritFed` stands in contrast to `TAWT`, which depends on computationally intensive hypergradient estimations, and `FedAdp`, which uses cosine similarity for weight calculation. In addition, we incorporate zero-order MD to enhance privacy. The key contributions of this paper include the development of `MeritFed`, provable convergence under mild assumptions, and the ability to utilize benefits from collaborating with clients having different but similar data.

However, our work has some limitations. Firstly, (in theory) `MeritFed` relies on the fact that the objective from the problem in Line 9 gives a good enough approximation of the expected risk $f$, which in some situations may require the availability of additional data on the target client to solve the problem (though in all of our experiments, it was not the case and `MeritFed` worked well even without additional data). Collecting and maintaining extra data may not always be practical or efficient. Secondly, the experiments used a limited number of clients and a dataset of moderate size. Extending `MeritFed` to large-scale FL with a substantial number of clients and massive datasets may pose scalability challenges. Addressing these limitations is part of our plan for future work.

Furthermore, `MeritFed` serves as a foundation for numerous extensions and enhancements. Future research can explore topics such as acceleration techniques, adaptive or scaled optimization methods (e.g., variants akin to `Adam` (Kingma & Ba, 2014)) on the server side, communication compression strategies, and the efficient implementation of similar collaborative learning approaches for all clients simultaneously. These directions will contribute to the continued development of FL methods, making them more efficient, robust, and applicable to a wide range of practical scenarios.

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

# A EXTENDED RELATED WORK

## A.1 RELATION TO TRANSFER LEARNING

While our approach resembles transfer learning (West et al., 2007), where a model trained on one dataset is then enhanced/fine-tuned on another related dataset, `MeritFed` differs significantly in both motivation and framework. Unlike transfer learning, which involves adapting a pre-trained model to new data, `MeritFed` enhances the training process itself. Transfer learning can be theoretically viewed as training with "better" initialization, while `MeritFed` decides on the fly what dataset to use and to what extent.

That is, `MeritFed` performs adaptive aggregation and benefits from clients having data with the same distribution. It promotes collaborative learning, which is particularly applicable in cross-silo federated learning (scenarios such as medical imaging).

Furthermore, in situations where datasets are unrelated, traditional transfer learning may not yield performance improvements. In contrast, `MeritFed` performs not worse than `SGD Ideal` under such conditions. Additionally, `MeritFed` provides robustness against Byzantine attacks, further distinguishing it from conventional transfer learning methods.

Exploring whether `MeritFed` can outperform transfer learning techniques in specific applications remains a valuable direction for future research but outside the scope of our work.

## A.2 PERSONALIZED FL BY GRAPH-BASED AGGREGATION

Another related direction in FL more accurately addresses client clustering by constructing a clients' relation graph. (Chen et al., 2022b) does a graph-based model aggregation (k-hop) based on an adaptively learned Graph Convolution Net (GCN). (Zhang et al., 2024) also uses GCN to perform graph-guided aggregation but focuses on recommendations. Both works lack theoretical analysis and require solving a subproblem (similar to BLO) of learning GCN at each iteration. This subproblem has a higher computation cost than MeritFed has for adaptive aggregation.

## A.3 WEIGHTS UPDATE FOR TAWT AND FEDADP

**TAWT.** A faithful implementation of `TAWT` (Chen et al., 2021a) would require a costly evaluation of the inverse of the Hessian matrix $\sum_{t=1}^{T} w_t \nabla^2 f(x^k)$ to calculate an approximation of hyper-gradient $g^k$. Then $g^k$ is supposed to be used to run one step of Mirror Descent (with step size $\eta^k$) to update the weights:

$$w_t^{k+1} = \frac{w_t^k \exp\{-\eta^k g_t^k\}}{\sum_{t'=1}^{T} w_{t'}^k \exp\{-\eta^k g_{t'}^k\}}. \tag{12}$$

In practice, (Chen et al., 2021a) advise bypassing this step by replacing the Hessian-inverse-weighted dissimilarity measure with a cosine-similarity-based measure, i.e., to approximate $g_t^k$ by $-c \times \mathcal{S}(\nabla f_0(x^k), \nabla f_t(x^k))$, where

$$\mathcal{S}(a, b) = \arccos \frac{\langle a, b \rangle}{\|a\| \|b\|}$$

denotes the cosine similarity between two vectors.

**FedAdp.** `FedAdp` (Wu & Wang, 2021) uses a similar update rule for weights, but it additionally uses a non-linear mapping function (*Gompertz function*)

$$\mathcal{G}(\xi) = \alpha \left(1 - e^{-e^{-\alpha\xi}}\right)$$

where $\xi$ is the *smoothed angle* in *radian*, $e$ denotes the exponential constant and $\alpha$ is a constant. By denoting $\mathcal{S}_t^k = \mathcal{S}(\nabla f_0(x^k), \nabla f_t(x^k))$ one can obtain `FedAdp` weights update rule in the form

$$w_t^k = \frac{e^{\mathcal{G}(\mathcal{S}_t^k)}}{\sum_{t'=1}^{n} e^{\mathcal{G}(\mathcal{S}_t^k)}}.$$

## A.4 MISSING APPROACHES FOR SOLVING AUXILIARY PROBLEM IN LINE 7

**Fresh Data.** Let us assume that the target client can obtain new samples from distribution $\mathcal{D}_1$ at any moment in time.

**Additional Validation Data.** Alternatively, one can assume that the target client has an additional validation dataset $\widehat{D}$ sampled from $\mathcal{D}_1$. Then, instead of function $f$ in Line 7, one can approximately minimize

$$\widehat{f}(x) = \frac{1}{|\widehat{D}|} \sum_{\xi \in \widehat{D}} f_\xi(x), \tag{13}$$

which under certain conditions provably approximates the original function $f(x)$ with any predefined accuracy if the dataset $\widehat{D}$ is sufficiently large (Shalev-Shwartz et al., 2009; Feldman & Vondrak, 2019). More precisely, the worst-case guarantees (e.g., (Liu & Tong, 2024)) imply that to guarantee $\mathbb{E}[f(\widehat{x}^*) - f(x^*)] \leq \delta$, where $\widehat{x}^* \in \arg\min_{x \in \mathbb{R}^d} \widehat{f}(x)$ and $x^* \in \arg\min_{x \in \mathbb{R}^d} f(x)$, the validation dataset should be of the size $|\widehat{\mathcal{D}}| \sim \max\{L/\mu, 1/\mu\delta\}$ under the assumption that $f_\xi(x)$ is $\mu$-strongly convex. However, as we observe in our experiments, MeritFed works well even with a relatively small size of the validation dataset for non-convex problems.

## B    PROOF OF THEOREM 1

We the theorem divide the proof into two parts: $K = 1$ and $K > 1$.

### B.1    NO LOCAL STEPS ($K = 1$)

**Theorem 3.** *Let Assumptions 1 holds. Then after $T$ iterations of* `MeritFed` *with $\gamma \leq \frac{1}{2L}$ outputs $x^t$, $t = 0, \cdots, T - 1$ such that*

$$\frac{1}{T} \sum_{t=0}^{T-1} \mathbb{E}\|\nabla f(x^t)\|^2 \leq \frac{2(f(x^0) - f(x^*))}{T\gamma} + \frac{2\sigma^2\gamma L}{G} + \frac{2\delta}{\gamma}, \tag{14}$$

*where $\delta$ is the accuracy of solving the problem in Line 7 and $G = |\mathcal{G}|$. Moreover if Assumption 3 additionally holds, then after $T$ iterations of* `MeritFed` *with $\gamma \leq \frac{1}{2L}$ outputs $x^T$ such that*

$$\mathbb{E}f(x^T) - f^* \leq (1 - \gamma\mu)^T (f(x^0) - f^*) + \frac{\sigma^2\gamma L}{\mu G} + \frac{\delta}{\gamma\mu}. \tag{15}$$

*Proof.* We write $g_i^t$ or simply $g_i$ instead of $g_i(x^t, \boldsymbol{\xi}_i^t)$ when there is no ambiguity. Then, the update rule in `MeritFed` can be written as

$$x^{t+1} = x^t - \gamma \sum_{i=0}^{n-1} w_i^{t+1} g_i(x^t),$$

where $w^{t+1}$ is an approximate solution of

$$\min_{w\Delta_1^n} f\left(x^t - \gamma \sum_{i=0}^{n-1} w_i g_i(x^t)\right)$$

that satisfies

$$\mathbb{E}\left[f(x^{t+1})|x^t, \boldsymbol{\xi}^t\right] - \min_w f\left(x^t - \gamma \sum_{i=0}^{n-1} w_i g_i(x^t)\right) \leq \delta.$$

By definition of the minimum, we have

$$\min_{w\in\Delta_1^n} f\left(x^t - \gamma \sum_{i=0}^{n-1} w_i g_i(x^t)\right) \leq f\left(x^t - \frac{\gamma}{G} \sum_{i\in\mathcal{G}} g_i(x^t)\right)$$

$$\overset{\text{(Lip)}}{\leq} f(x^t) - \frac{\gamma}{G}\left\langle \nabla f(x^t), \sum_{i\in\mathcal{G}} g_i(x^t) \right\rangle + \frac{L\gamma^2}{2}\left\|\frac{1}{G} \sum_{i\in\mathcal{G}} g_i(x^t)\right\|^2$$

$$\leq f(x^t) - \frac{\gamma}{G}\left\langle \nabla f(x^t), \sum_{i\in\mathcal{G}} g_i(x^t) \right\rangle + \gamma^2 L\left\|\nabla f(x^t) - \frac{1}{G} \sum_{i\in\mathcal{G}} g_i(x^t)\right\|^2$$

$$+ \gamma^2 L\|\nabla f(x^t)\|^2.$$

The last two inequalities imply

$$\mathbb{E}\left[f(x^{t+1})|x^t, \boldsymbol{\xi}^t\right]$$

$$\leq f(x^t) - \frac{\gamma}{G}\left\langle \nabla f(x^t), \sum_{i\in\mathcal{G}} g_i(x^t) \right\rangle + \gamma^2 L\left\|\nabla f(x^t) - \frac{\sum_{i\in\mathcal{G}} g_i(x^t)}{G}\right\|^2$$

$$+ \gamma^2 L\|\nabla f(x^t)\|^2 + \delta.$$

Taking the full expectation we get

$$
\begin{aligned}
\mathbb{E}[f(x^{t+1})] \quad \leq \quad & \mathbb{E}[f(x^t)] - \gamma(1 - \gamma L)\mathbb{E}\left[\|\nabla f(x^t)\|^2\right] \\
& + \gamma^2 L \mathbb{E}\left[\left\|\nabla f(x^t) - \frac{\sum_{i \in \mathcal{G}} g_i(x^t)}{G}\right\|^2\right] + \delta \\
\overset{\gamma \leq \frac{1}{2L}}{\leq} \quad & \mathbb{E}[f(x^t)] - \frac{\gamma}{2}\mathbb{E}\left[\|\nabla f(x^t)\|^2\right] + \frac{\gamma^2 L}{G^2}\sum_{i \in \mathcal{G}}\mathbb{E}\left[\|\nabla f(x^t) - g_i(x^t)\|^2\right] + \delta \\
\overset{(10)}{\leq} \quad & \mathbb{E}[f(x^t)] - \frac{\gamma}{2}\mathbb{E}\left[\|\nabla f(x^t)\|^2\right] + \frac{\gamma^2 L \sigma^2}{G} + \delta.
\end{aligned}
\tag{16}
$$

The above is equivalent to

$$
\frac{\gamma}{2}\mathbb{E}\|\nabla f(x^t)\|^2 \leq \mathbb{E}f(x^t) - \mathbb{E}f(x^{t+1}) + \frac{\sigma^2 \gamma^2 L}{G} + \delta,
$$

which concludes the first part of the proof.

Next, summing the inequality for $t \in \{0, 1, \dots, T-1\}$ leads to

$$
\begin{aligned}
\frac{1}{T}\sum_{t=0}^{T-1}\mathbb{E}\|\nabla f(x^t)\|^2 \quad & \leq \quad \frac{2(f(x^0) - \mathbb{E}f(x^T))}{T\gamma} + \frac{2\sigma^2 \gamma L}{G} + \frac{2\delta}{\gamma} \\
& \leq \quad \frac{2(f(x^0) - f(x^*))}{T\gamma} + \frac{2\sigma^2 \gamma L}{G} + \frac{2\delta}{\gamma}.
\end{aligned}
$$

Combining (16) with (PL) gives

$$
\gamma\mu\mathbb{E}[f(x^t) - f^*] \leq \frac{\gamma}{2}\mathbb{E}\left[\|\nabla f(x^t)\|^2\right] \leq \mathbb{E}[f(x^t)] - \mathbb{E}[f(x^{t+1})] + \frac{\gamma^2 L \sigma^2}{G} + \delta,
$$

or equivalently

$$
\mathbb{E}[f(x^{t+1})] - f^* \quad \leq \quad (1 - \gamma\mu)\mathbb{E}[f(x^t) - f^*] + \frac{\gamma^2 L \sigma^2}{G} + \delta.
$$

The above unrolls as

$$
\begin{aligned}
\mathbb{E}f(x^T) - f^* \quad & \leq \quad (1 - \gamma\mu)^T(f(x^0) - f^*) + \left(\frac{\sigma_G^2 \gamma^2 L}{G} + \delta\right)\sum_{t=0}^{T-1}(1 - \gamma\mu)^t \\
& \leq \quad (1 - \gamma\mu)^T(f(x^0) - f^*) + \left(\frac{\sigma_G^2 \gamma^2 L}{G} + \delta\right)\sum_{t=0}^{\infty}(1 - \gamma\mu)^t \\
& \leq \quad (1 - \gamma\mu)^T(f(x^0) - f^*) + \frac{\gamma L \sigma_G^2}{\mu G} + \frac{\delta}{\gamma\mu},
\end{aligned}
$$

which is the result of the theorem (15). $\qquad\square$

## B.2  LOCAL STEPS ($K > 1$)

The derivation is based on Reddi et al. (2020).

**Lemma 1.** *For independent, mean 0 random variables $z_1, \ldots, z_r$, we have*

$$\mathbb{E}\left[\|z_1 + \ldots + z_r\|^2\right] = \mathbb{E}\left[\|z_1\|^2 + \ldots + \|z_r\|^2\right].$$

**Lemma 2.** *For any step-size satisfying $\gamma_l \leq \frac{1}{3LK}$, we can bound the drift for any $k \in \{0, \cdots, K-1\}$ as*

$$\frac{1}{G}\sum_{i=1}^{G} \mathbb{E}\|x_{i,k}^t - x_t\|^2 \leq 5K\gamma_l^2\sigma_G^2 + 20K^2\gamma_l^2\mathbb{E}[\|\nabla f(x_t)))\|^2]. \tag{17}$$

*Proof.* The result trivially holds for $k = 1$ since $x_{i,0}^t = x_t$ for all $i \in [m]$. We now turn our attention to the case where $k \geq 1$. To prove the above result, we observe that for any client $i \in [m]$ and $k \in [K]$,

$$\mathbb{E}\|x_{i,k}^t - x_t\|^2 = \mathbb{E}\left\|x_{i,k-1}^t - x_t - \gamma_l g_{i,k-1}^t\right\|^2$$

$$\leq \mathbb{E}\left\|x_{i,k-1}^t - x_t - \gamma_l(g_{i,k-1}^t - \nabla f(x_{i,k-1}^t) + \nabla f(x_{i,k-1}^t) - \nabla f(x_t) + \nabla f(x_t))\right\|^2$$

$$\leq \left(1 + \frac{1}{2K-1}\right)\mathbb{E}\left\|x_{i,k-1}^t - x_t\right\|^2 + \mathbb{E}\left\|\gamma_l(g_{i,k-1}^t - \nabla f(x_{i,k-1}^t))\right\|^2$$

$$+ 4K\mathbb{E}[\|\gamma_l(\nabla f(x_{i,k-1}^t) - \nabla f(x_t))\|^2] + 4K\mathbb{E}[\|\gamma_l \nabla f(x_t)))\|^2]$$

The first inequality uses the fact that $g_{k-1,i}^t$ is an unbiased estimator of $\nabla f(x_{i,k-1}^t)$ and Lemma 1. The above quantity can be further bounded by the following:

$$\mathbb{E}\|x_{i,k}^t - x_t\|^2 \leq \left(1 + \frac{1}{2K-1}\right)\mathbb{E}\|x_{i,k-1}^t - x_t\|^2 + \gamma_l^2\sigma_G^2 + 4K\gamma_l^2\mathbb{E}\|L(x_{i,k-1}^t - x_t)\|^2$$

$$+ 4K\mathbb{E}[\|\gamma_l \nabla f(x_t)))\|^2]$$

$$= \left(1 + \frac{1}{2K-1} + 4K\gamma_l^2 L^2\right)\mathbb{E}\|(x_{i,k-1}^t - x_t)\|^2 + \gamma_l^2\sigma_G^2$$

$$+ 4K\gamma_l^2\mathbb{E}[\|\nabla f(x_t)))\|^2]$$

$$= \left(1 + \frac{1}{K-1}\right)\mathbb{E}\|(x_{i,k-1}^t - x_t)\|^2 + \gamma_l^2\sigma_G^2$$

$$+ 4K\gamma_l^2\mathbb{E}[\|\nabla f(x_t)))\|^2]$$

Here, the first inequality follows from Assumption 1, and the last one from $4K\gamma_l^2 L^2 \leq \frac{4}{9K}$ the following chain:

$$\frac{1}{2K-1} = \frac{1}{2K-1} \pm \frac{1}{K-1} = \frac{1}{K-1} - \frac{K}{(2K-1)(K-1)} \leq \frac{1}{K-1} - \frac{4}{9K}.$$

Averaging over the clients $i$, we obtain the following:

$$\frac{1}{G}\sum_{i=1}^{G}\mathbb{E}\|x_{i,k}^t - x_t\|^2 \leq \left(1 + \frac{1}{K-1}\right)\frac{1}{G}\sum_{i=1}^{G}\mathbb{E}\|x_{i,k-1}^t - x_t\|^2 + \gamma_l^2\sigma_G^2$$

$$+ 4K\gamma_l^2\mathbb{E}[\|\nabla f(x_t)))\|^2]$$

Unrolling the recursion, we obtain the following:

$$\frac{1}{G}\sum_{i=1}^{G}\mathbb{E}\|x_{i,k}^t - x_t\|^2 \leq \sum_{p=0}^{k-1}\left(1 + \frac{1}{K-1}\right)^p \left[\gamma_l^2\sigma_G^2 + 4K\gamma_l^2\mathbb{E}[\|\nabla f(x_t)))\|^2]\right]$$

$$\leq (K-1) \times \left[\left(1 + \frac{1}{K-1}\right)^K - 1\right] \times \left[\gamma_l^2\sigma_G^2 + 4K\gamma_l^2\mathbb{E}[\|\nabla f(x_t)))\|^2]\right]$$

$$\leq 5K\gamma_l^2\sigma_G^2 + 20K^2\gamma_l^2\mathbb{E}[\|\nabla f(x_t)))\|^2],$$

concluding the proof of Lemma 2. The last inequality uses the fact that $(1 + \frac{1}{K-1})^K \leq 5$ for $K > 1$. $\qquad\square$

**Theorem 4.** *Let Assumptions 1 holds. Then after $T$ iterations of* `MeritFed` *with $\gamma = 2$, $\gamma_l \leq \frac{1}{12LK}$ outputs $x^t$, $t = 0, \cdots, T - 1$ such that*

$$\frac{1}{T} \sum_{t=0}^{T-1} \mathbb{E}\|\nabla f(x^t)\|^2 \leq \frac{4\big(f(x^0) - \mathbb{E}f(x^T)\big)}{\gamma_l K T} + 24\gamma_l^2 K L^2 \sigma_G^2 + \frac{32\gamma_l L \sigma_G^2}{G} + \frac{4\delta}{\gamma_l K},$$

*where $\delta$ is the accuracy of solving the problem in Line 7 and $G = |\mathcal{G}|$. Moreover if Assumption 3 additionally holds, then after $T$ iterations of* `MeritFed` *outputs $x^T$ such that*

$$\mathbb{E}[f(x^T) - f^*] \leq \Big(1 - \frac{\mu\gamma_l K}{2}\Big)^T [f(x^0) - f^*] + \frac{12\gamma_l^2 K L^2 \sigma_G^2}{\mu} + \frac{16\gamma_l L \sigma_G^2}{\mu G} + \frac{2\delta}{\mu\gamma_l K}.$$

*Proof.* We write $g_i^t$ or simply $g_i$ instead of $g_i(x^t, \boldsymbol{\xi}_i^t)$ when there is no ambiguity. Then, the update rule in `MeritFed` can be written as

$$x^{t+1} = x^t + \gamma \sum_{i=0}^{n-1} w_i^{t+1} \Delta_i^t,$$

where $w^{t+1}$ is an approximate solution of

$$\min_{w\Delta_1^n} f\Big(x^t + \gamma \sum_{i=0}^{n-1} w_i \Delta_i^t\Big)$$

that satisfies

$$\mathbb{E}\big[f\big(x^{t+1}\big)|x^t, \boldsymbol{\xi}^t\big] - \min_w f\Big(x^t + \gamma \sum_{i=0}^{n-1} w_i \Delta_i^t\Big) \leq \delta.$$

By definition of the minimum, we have

$$\min_{w\in\Delta_1^n} f\Big(x^t + \gamma \sum_{i=0}^{n-1} w_i \Delta_i^t\Big) \leq f\Big(x^t + \frac{\gamma}{G} \sum_{i\in\mathcal{G}} \Delta_i^t\Big)$$

$$\overset{\text{(Lip)}}{\leq} f\big(x^t\big) + \frac{\gamma}{G}\Big\langle \nabla f(x^t), \sum_{i\in\mathcal{G}} \Delta_i^t \Big\rangle + \frac{L\gamma^2}{2}\Big\|\frac{1}{G} \sum_{i\in\mathcal{G}} \Delta_i^t\Big\|^2$$

$$= f\big(x^t\big) + \frac{\gamma}{G}\Big\langle \nabla f(x^t), \sum_{i\in\mathcal{G}} \Delta_i^t + \gamma_l KG\nabla f(x^t) \Big\rangle - \gamma\gamma_l K\|\nabla f(x^t)\|^2 + \frac{L\gamma^2}{2G^2}\Big\|\sum_{i\in\mathcal{G}} \Delta_i^t\Big\|^2$$

Next we bound

$$\mathbb{E}\Big\langle \nabla f\big(x^t\big), \sum_{i\in\mathcal{G}} \Delta_i^t + \gamma_l KG\nabla f(x^t) \Big\rangle = \mathbb{E}\Big\langle \nabla f\big(x^t\big), \gamma_l KG\nabla f(x^t) - \sum_{i\in\mathcal{G}} \sum_{k=0}^{K-1} \gamma_l g_{i,k}^t \Big\rangle$$

$$\leq \frac{\gamma_l KG}{2}\|\nabla f(x^t)\|^2 + \frac{1}{2\gamma_l KG}\Big\|\gamma_l \sum_{i\in\mathcal{G}} \sum_{k=0}^{K-1} \big(\nabla f(x^t) - \nabla f(x_{i,k}^t)\big)\Big\|^2$$

$$\leq \frac{\gamma_l KG}{2}\|\nabla f(x^t)\|^2 + \frac{L^2\gamma_l}{2} \sum_{i\in\mathcal{G}} \sum_{k=0}^{K-1} \|x^t - x_{i,k}^t\|^2,$$

where we used unbiasedness given by Assumption 1, and

$$\left\|\sum_{i\in\mathcal{G}}\Delta_i^t\right\|^2 = \left\|\sum_{i\in\mathcal{G}}\sum_{k=0}^{K-1}\gamma_l g_{i,k}^t\right\|^2$$

$$\leq 2\left\|\sum_{i\in\mathcal{G}}\sum_{k=0}^{K-1}\gamma_l g_{i,k}^t - \gamma_l KG\nabla f(x^t)\right\|^2 + 2\left\|\gamma_l KG\nabla f(x^t)\right\|^2$$

$$\leq 2\left\|\gamma_l\sum_{i\in\mathcal{G}}\sum_{k=0}^{K-1}\left(g_{i,k}^t - \nabla f(x_{i,k}^t)\right) + \left(\nabla f(x_{i,k}^t) - \nabla f(x^t)\right)\right\|^2 + 2\left\|\gamma_l KG\nabla f(x^t)\right\|^2$$

$$\leq 4\gamma_l^2 KG\sum_{i\in\mathcal{G}}\sum_{k=0}^{K-1}\left\|g_{i,k}^t - \nabla f(x_{i,k}^t)\right\|^2 + 4\gamma_l^2 KG\sum_{i\in\mathcal{G}}\sum_{k=0}^{K-1}\left\|\nabla f(x_{i,k}^t) - \nabla f(x^t)\right\|^2$$

$$+ 2\left\|\gamma_l KG\nabla f(x^t)\right\|^2$$

$$\leq 4\gamma_l^2 KG\sum_{i\in\mathcal{G}}\sum_{k=0}^{K-1}\left\|g_{i,k}^t - \nabla f(x_{i,k}^t)\right\|^2 + 4\gamma_l^2 KG\sum_{i\in\mathcal{G}}\sum_{k=0}^{K-1}\left\|\nabla f(x_{i,k}^t) - \nabla f(x^t)\right\|^2$$

$$+ 2\left\|\gamma_l KG\nabla f(x^t)\right\|^2$$

$$\leq 4\gamma_l^2 KG\sum_{i\in\mathcal{G}}\sum_{k=0}^{K-1}\left\|g_{i,k}^t - \nabla f(x_{i,k}^t)\right\|^2 + 4\gamma_l^2 KGL^2\sum_{i\in\mathcal{G}}\sum_{k=0}^{K-1}\left\|x_{i,k}^t - x^t\right\|^2$$

$$+ 2\left\|\gamma_l KG\nabla f(x^t)\right\|^2.$$

Taking an expectation we obtain

$$\mathbb{E}\left\|\sum_{i\in\mathcal{G}}\Delta_i^t\right\|^2 \leq 4\gamma_l^2 KG\sigma_G^2 + 4\gamma_l^2 KGL^2\sum_{i\in\mathcal{G}}\sum_{k=0}^{K-1}\mathbb{E}\left\|x_{i,k}^t - x^t\right\|^2 + 2\left\|\gamma_l KG\nabla f(x^t)\right\|^2.$$

The inequalities above imply

$$\mathbb{E}f\left(x^{t+1}\right)$$

$$\leq \mathbb{E}f\left(x^t\right) - \gamma\gamma_l K\mathbb{E}\|\nabla f(x^t)\|^2 + \frac{\gamma_l\gamma K}{2}\mathbb{E}\|\nabla f(x^t)\|^2 + \frac{\gamma_l\gamma L^2}{2G}\sum_{i\in\mathcal{G}}\sum_{k=0}^{K-1}\mathbb{E}\|x^t - x_{i,k}^t\|^2$$

$$+ \frac{2\gamma_l^2\gamma^2 KL\sigma_G^2}{G} + \frac{2\gamma_l^2 KL^3}{G}\sum_{i\in\mathcal{G}}\sum_{k=0}^{K-1}\mathbb{E}\|x_{i,k}^t - x^t\|^2 + L\gamma_l^2\gamma^2 K^2\mathbb{E}\|\nabla f(x^t)\|^2 + \delta$$

$$\overset{\text{Lemma 2}}{\leq} \mathbb{E}f\left(x^t\right) - \frac{\gamma_l\gamma K}{2}\mathbb{E}\|\nabla f(x^t)\|^2 + \frac{\gamma_l\gamma L^2}{2}K\left(5K\gamma_l^2\sigma_G^2 + 20K^2\gamma_l^2\mathbb{E}\|\nabla f(x_t)\|^2\right)$$

$$+ \frac{2\gamma_l^2\gamma^2 KL\sigma_G^2}{G} + 2\gamma_l^2 K^2 L^3\left(5K\gamma_l^2\sigma_G^2 + 20K^2\gamma_l^2\mathbb{E}\|\nabla f(x_t)\|^2\right)$$

$$+ L\gamma_l^2\gamma^2 K^2\|\nabla f(x^t)\|^2 + \delta$$

$$= \mathbb{E}f\left(x^t\right) + \frac{\gamma_l K}{2}\left\{\gamma L^2 20K^2\gamma_l^2 + 80\gamma_l^3 K^3 L^3 + 2L\gamma^2\gamma_l K - \gamma\right\}\mathbb{E}\|\nabla f(x^t)\|^2$$

$$+ \frac{5K^2\gamma_l^3\sigma_G^2\gamma L^2}{2} + \frac{2\gamma_l^2\gamma^2 KL\sigma_G^2}{G} + 10\gamma_l^4 K^3 L^3\sigma_G^2 + \delta$$

Setting $\gamma_l \leq \frac{1}{12LK}$, $\gamma = 2$ we obtain

$$\mathbb{E}f\left(x^{t+1}\right)$$

$$\leq \quad \mathbb{E}f\left(x^t\right) - \frac{\gamma_l K}{4}\mathbb{E}\|\nabla f(x^t)\|^2 + 10\gamma_l^4 K^3 L^3 \sigma_G^2 + 5\gamma_l^3 K^2 L^2 \sigma_G^2 + \frac{8\gamma_l^2 KL\sigma_G^2}{G} + \delta$$

$$\leq \quad \mathbb{E}f\left(x^t\right) - \frac{\gamma_l K}{4}\mathbb{E}\|\nabla f(x^t)\|^2 + \frac{10}{12}\gamma_l^3 K^2 L^2 \sigma_G^2 + 5\gamma_l^3 K^2 L^2 \sigma_G^2 + \frac{8\gamma_l^2 KL\sigma_G^2}{G} + \delta$$

$$\leq \quad \mathbb{E}f\left(x^t\right) - \frac{\gamma_l K}{4}\mathbb{E}\|\nabla f(x^t)\|^2 + 6\gamma_l^3 K^2 L^2 \sigma_G^2 + \frac{8\gamma_l^2 KL\sigma_G^2}{G} + \delta$$

The above is equivalent to

$$\frac{\gamma_l K}{4}\mathbb{E}\|\nabla f(x^t)\|^2 \leq \mathbb{E}f\left(x^t\right) - \mathbb{E}f\left(x^{t+1}\right) + 6\gamma_l^3 K^2 L^2 \sigma_G^2 + \frac{8\gamma_l^2 KL\sigma_G^2}{G} + \delta,$$

which concludes the first part of the proof.

Next, summing the inequality for $t \in \{0, 1, \ldots, T-1\}$ leads to

$$\frac{1}{T}\sum_{t=0}^{T-1}\mathbb{E}\left\|\nabla f\left(x^t\right)\right\|^2$$

$$\leq \quad \frac{4\left(f\left(x^0\right) - \mathbb{E}f\left(x^T\right)\right)}{\gamma_l K T} + 24\gamma_l^2 K L^2 \sigma_G^2 + \frac{32\gamma_l L\sigma_G^2}{G} + \frac{4\delta}{\gamma_l K}.$$

Combining (16) with (PL) gives

$$\frac{\mu\gamma_l K}{2}\mathbb{E}[f(x^t) - f^*] \leq \frac{\gamma_l K}{4}\mathbb{E}\left[\|\nabla f(x^t)\|^2\right]$$

$$\leq \mathbb{E}[f(x^t)] - \mathbb{E}[f(x^{t+1})] + 6\gamma_l^3 K^2 L^2 \sigma_G^2 + \frac{8\gamma_l^2 KL\sigma_G^2}{G} + \delta,$$

or equivalently

$$\mathbb{E}[f(x^{t+1}) - f^*] \leq \left(1 - \frac{\mu\gamma_l K}{2}\right)\mathbb{E}[f(x^t) - f^*] + 6\gamma_l^3 K^2 L^2 \sigma_G^2 + \frac{8\gamma_l^2 KL\sigma_G^2}{G} + \delta.$$

The above unrolls as

$$\mathbb{E}[f\left(x^T\right) - f^*]$$

$$\leq \quad \left(1 - \frac{\mu\gamma_l K}{2}\right)^T \left(f\left(x^0\right) - f^*\right)$$

$$+ \left(+6\gamma_l^3 K^2 L^2 \sigma_G^2 + \frac{8\gamma_l^2 KL\sigma_G^2}{G} + \delta\right)\sum_{t=0}^{T-1}\left(1 - \frac{\mu\gamma_l K}{2}\right)^t$$

$$\leq \quad \left(1 - \frac{\mu\gamma_l K}{2}\right)^T \left(f\left(x^0\right) - f^*\right)$$

$$+ \left(+6\gamma_l^3 K^2 L^2 \sigma_G^2 + \frac{8\gamma_l^2 KL\sigma_G^2}{G} + \delta\right)\sum_{t=0}^{\infty}\left(1 - \frac{\mu\gamma_l K}{2}\right)^t$$

$$\leq \quad \left(1 - \frac{\mu\gamma_l K}{2}\right)^T \left(f\left(x^0\right) - f^*\right) + \frac{12\gamma_l^2 K L^2 \sigma_G^2}{\mu} + \frac{16\gamma_l L\sigma_G^2}{\mu G} + \frac{2\delta}{\mu\gamma_l K}$$

which is the result of the theorem (15). $\qquad\square$

## C  PROOF OF THEOREM 2

**Theorem 5.** *Let Assumptions 1 and 2 hold with $G = |\mathcal{G}| > 0$, $F = |\mathcal{F}| > 0$, $\nu \leq \frac{G}{F}$. Then after $T$ iterations of* `MeritFed` *with $\gamma \leq \frac{1}{8L}$ outputs $x^t$, $t = 0, \cdots, T-1$ such that*

$$\frac{1}{T}\sum_{t=0}^{T-1}\mathbb{E}\big\|\nabla f(x^t)\big\|^2 \leq \frac{4\big(f(x^0) - \mathbb{E}f(x^T)\big)}{\gamma T} + \frac{8\gamma LG\sigma_{\mathcal{G}}^2}{(G+F)^2} + \frac{8\gamma LF\sigma_{\mathcal{F}}^2}{(G+F)^2} + \frac{2\rho^2 F}{G+F} + \frac{4\delta}{\gamma}, \quad (18)$$

*where $\delta$ is the accuracy of solving the problem in Line 7. Moreover if Assumption 3 additionally holds, then after $T$ iterations of* `MeritFed` *with $\gamma \leq \frac{1}{8L}$ outputs $x^T$ such that*

$$\mathbb{E}f(x^T) - f^* \leq (1 - \gamma\mu)^T\big(f(x^0) - f^*\big) + \frac{4\gamma LG\sigma_{\mathcal{G}}^2}{\mu(G+F)^2} + \frac{4\gamma LF\sigma_{\mathcal{F}}^2}{\mu(G+F)^2} + \frac{\rho^2}{\mu}\frac{F}{G+F} + \frac{2\delta}{\gamma\mu}. \quad (19)$$

*Proof.* We write $g_i^t$ or simply $g_i$ instead of $g_i(x^t, \boldsymbol{\xi}_i^t)$ when there is no ambiguity. Then, the update rule of `MeritFed` can be written as

$$x^{t+1} = x^t - \gamma\sum_{i=0}^{n-1} w_i^{t+1} g_i(x^t),$$

where $w^{t+1}$ is an approximate solution of

$$\min_{w\Delta_1^n} f\left(x^t - \gamma\sum_{i=0}^{n-1} w_i g_i(x^t)\right)$$

that satisfies

$$\mathbb{E}\big[f(x^{t+1})|x^t, \boldsymbol{\xi}^t\big] - \min_w f\left(x^t - \gamma\sum_{i=0}^{n-1} w_i g_i(x^t)\right) \leq \delta.$$

By definition of the minimum, we have

$$\min_{w\in\Delta_1^n} f\left(x^t - \gamma\sum_{i=0}^{n-1} w_i g_i(x^t)\right) \leq f\left(x^t - \frac{\gamma}{G+F}\sum_{i\in\mathcal{G}\cup\mathcal{F}} g_i(x^t)\right)$$

$$\overset{\text{(Lip)}}{\leq} f(x^t) - \frac{\gamma}{G+F}\left\langle\nabla f(x^t), \sum_{i\in\mathcal{G}\cup\mathcal{F}} g_i(x^t)\right\rangle + \frac{L\gamma^2}{2}\left\|\frac{1}{G+F}\sum_{i\in\mathcal{G}\cup\mathcal{F}} g_i(x^t)\right\|^2$$

$$\leq f(x^t) + \frac{2\gamma^2 LG^2}{(G+F)^2}\big\|\nabla f(x^t)\big\|^2 + \frac{2\gamma^2 L}{(G+F)^2}\left\|\sum_{i\in\mathcal{F}}\nabla f_i(x^t)\right\|^2$$

$$-\frac{\gamma}{G+F}\left\langle\nabla f(x^t), \sum_{i\in\mathcal{G}} g_i(x^t)\right\rangle + \frac{2\gamma^2 L}{(G+F)^2}\left\|\sum_{i\in\mathcal{G}}\nabla f(x^t) - g_i(x^t)\right\|^2$$

$$-\frac{\gamma}{G+F}\left\langle\nabla f(x^t), \sum_{i\in\mathcal{F}} g_i(x^t)\right\rangle + \frac{2\gamma^2 L}{(G+F)^2}\left\|\sum_{i\in\mathcal{F}}\nabla f_i(x^t) - g_i(x^t)\right\|^2.$$

The last two inequalities imply

$$\mathbb{E}\big[f(x^{t+1})|x^t, \boldsymbol{\xi}^t\big]$$

$$\leq f(x^t) + \frac{2\gamma^2 LG^2}{(G+F)^2}\big\|\nabla f(x^t)\big\|^2 + \frac{2\gamma^2 L}{(G+F)^2}\left\|\sum_{i\in\mathcal{F}}\nabla f_i(x^t)\right\|^2 + \delta.$$

$$-\frac{\gamma}{G+F}\left\langle\nabla f(x^t), \sum_{i\in\mathcal{G}} g_i(x^t)\right\rangle + \frac{2\gamma^2 L}{(G+F)^2}\left\|\sum_{i\in\mathcal{G}}\nabla f(x^t) - g_i(x^t)\right\|^2$$

$$-\frac{\gamma}{G+F}\left\langle\nabla f(x^t), \sum_{i\in\mathcal{F}} g_i(x^t)\right\rangle + \frac{2\gamma^2 L}{(G+F)^2}\left\|\sum_{i\in\mathcal{F}}\nabla f_i(x^t) - g_i(x^t)\right\|^2$$

Taking an expectation conditioned on $x^t$ we get

$$\mathbb{E}[f(x^{t+1})|x^t]$$

$$\leq \quad f(x^t) - \frac{\gamma}{2}\left(1 - \frac{4\gamma LG^2}{(G+F)^2}\right)\|\nabla f(x^t)\|^2 + \frac{\gamma(F-G)}{2(G+F)}\|\nabla f(x^t)\|^2 + \frac{2\gamma^2 LG\sigma_{\mathcal{G}}^2}{(G+F)^2}$$

$$- \frac{\gamma}{G+F}\left\langle \nabla f(x^t), \sum_{i\in\mathcal{F}}\nabla f_i(x^t)\right\rangle + \frac{2\gamma^2 L}{(G+F)^2}\left\|\sum_{i\in\mathcal{F}}\nabla f_i(x^t)\right\|^2 + \frac{2\gamma^2 LF\sigma_{\mathcal{F}}^2}{(G+F)^2} + \delta$$

$$\overset{\gamma\leq\frac{(G+F)^2}{8LG^2}}{\leq} \quad f(x^t) - \frac{\gamma}{4}\|\nabla f(x^t)\|^2 + \frac{2\gamma^2 LG\sigma_{\mathcal{G}}^2}{(G+F)^2} + \frac{2\gamma^2 LF\sigma_{\mathcal{F}}^2}{(G+F)^2} + \delta$$

$$- \frac{\gamma}{G+F}\left\langle \nabla f(x^t), \sum_{i\in\mathcal{F}}\nabla f_i(x^t)\right\rangle + \frac{2\gamma^2 L}{(G+F)^2}\left\|\sum_{i\in\mathcal{F}}\nabla f_i(x^t)\right\|^2 \qquad (20)$$

$$+ \frac{\gamma(F-G)}{2(G+F)}\|\nabla f(x^t)\|^2$$

$$= \quad f(x^t) - \frac{\gamma}{4}\|\nabla f(x^t)\|^2 + \frac{2\gamma^2 LG\sigma_{\mathcal{G}}^2}{(G+F)^2} + \frac{2\gamma^2 LF\sigma_{\mathcal{F}}^2}{(G+F)^2} + \delta$$

$$+ \frac{1}{2}\frac{\gamma F}{G+F}\left\|\frac{1}{F}\sum_{i\in\mathcal{F}}\nabla f_i(x^t) - \nabla f(x^t)\right\|^2 - \frac{1}{2}\frac{\gamma G}{G+F}\|\nabla f(x^t)\|^2$$

$$- \frac{1}{2}\frac{\gamma F}{G+F}\left\|\frac{1}{F}\sum_{i\in\mathcal{F}}\nabla f_i(x^t)\right\|^2 + \frac{2\gamma^2 L}{(G+F)^2}\left\|\sum_{i\in\mathcal{F}}\nabla f_i(x^t)\right\|^2$$

$$\overset{(11)}{\leq} \quad f(x^t) - \frac{\gamma}{4}\|\nabla f(x^t)\|^2 + \frac{2\gamma^2 LG\sigma_{\mathcal{G}}^2}{(G+F)^2} + \frac{2\gamma^2 LF\sigma_{\mathcal{F}}^2}{(G+F)^2} + \delta + \frac{\rho^2}{2}\frac{\gamma F}{G+F}$$

$$+ \frac{\nu}{2}\frac{\gamma F}{G+F}\|\nabla f(x^t)\|^2 - \frac{1}{2}\frac{\gamma G}{G+F}\|\nabla f(x^t)\|^2$$

$$- \frac{1}{2}\frac{\gamma F}{G+F}\left(1 - \frac{2\gamma LF}{G+F}\right)\left\|\frac{1}{F}\sum_{i\in\mathcal{F}}\nabla f_i(x^t)\right\|^2 \qquad (21)$$

Next, since $\nu \leq \frac{G}{F}$ and $\gamma \leq \frac{1}{8L} \leq \frac{(G+F)}{4LF}$, we can take the full expectation from (21) and get

$$\frac{\gamma}{4}\mathbb{E}\left\|\nabla f(x^t)\right\|^2 \leq \mathbb{E}f(x^t) - \mathbb{E}f(x^{t+1}) + \frac{2\gamma^2 LG\sigma_{\mathcal{G}}^2}{(G+F)^2} + \frac{2\gamma^2 LF\sigma_{\mathcal{F}}^2}{(G+F)^2} + \frac{\rho^2}{2}\frac{\gamma F}{G+F} + \delta,$$

Summing up the above inequality for $t\in\{0,1,\ldots,T-1\}$, we derive

$$\frac{1}{T}\sum_{t=0}^{T-1}\mathbb{E}\left\|\nabla f(x^t)\right\|^2 \leq \frac{4(f(x^0) - \mathbb{E}f(x^T))}{\gamma T} + \frac{8\gamma LG\sigma_{\mathcal{G}}^2}{(G+F)^2} + \frac{8\gamma LF\sigma_{\mathcal{F}}^2}{(G+F)^2} + \frac{2\rho^2 F}{G+F} + \frac{4\delta}{\gamma},$$

which gives the first part of the result.

Next, if Assumption 3 holds, we combine (22) with (PL):

$$\mathbb{E}[f(x^{t+1}) - f^*] \leq (1-\gamma\mu)\mathbb{E}[f(x^t) - f^*] + \frac{4\gamma^2 LG\sigma_{\mathcal{G}}^2}{(G+F)^2} + \frac{4\gamma^2 LF\sigma_{\mathcal{F}}^2}{(G+F)^2} + \frac{\gamma\rho^2 F}{G+F} + 2\delta.$$

Unrolling the above recurrence, we obtain

$$\mathbb{E}f(x^T) - f^* \leq (1-\gamma\mu)^T(f(x^0) - f^*) + \frac{4\gamma LG\sigma_{\mathcal{G}}^2}{\mu(G+F)^2} + \frac{4\gamma LF\sigma_{\mathcal{F}}^2}{\mu(G+F)^2} + \frac{\rho^2}{\mu}\frac{F}{G+F} + \frac{2\delta}{\gamma\mu}.$$

$$\square$$

**Theorem 6.** *Let Assumptions 1 and 2 hold with $G = |\mathcal{G}|$, $F = |\mathcal{F}|$, $\nu \leq \frac{G}{F}$. Then after $T$ iterations of* `MeritFed` *with $\gamma \leq \frac{1}{8L}$ outputs $x^t$, $t = 0, \cdots, T-1$ such that*

$$\frac{1}{T} \sum_{t=0}^{T-1} \mathbb{E}\big\|\nabla f\big(x^t\big)\big\|^2 \leq \frac{4\big(f\big(x^0\big) - f(x^*)\big)}{T\gamma}$$

$$+ \quad \min\left\{\frac{2\sigma^2\gamma L}{G} + \frac{2\delta}{\gamma}, \quad \frac{8\gamma LG\sigma_{\mathcal{G}}^2}{(G+F)^2} + \frac{8\gamma LF\sigma_{\mathcal{F}}^2}{(G+F)^2} + \frac{2\rho^2 F}{G+F} + \frac{4\delta}{\gamma}\right\}, \tag{22}$$

*where $\delta$ is the accuracy of solving the problem in Line 7. Moreover if Assumption 3 additionally holds, then after $T$ iterations of* `MeritFed` *with $\gamma \leq \frac{1}{8L}$ outputs $x^T$ such that*

$$\mathbb{E}f\big(x^T\big) - f^* \leq (1 - \gamma\mu)^T \big(f\big(x^0\big) - f^*\big) + \tag{23}$$

$$\min\left\{\frac{\sigma^2\gamma L}{\mu G} + \frac{\delta}{\gamma\mu}, \quad \frac{4\gamma LG\sigma_{\mathcal{G}}^2}{\mu(G+F)^2} + \frac{4\gamma LF\sigma_{\mathcal{F}}^2}{\mu(G+F)^2} + \frac{\rho^2}{\mu}\frac{F}{G+F} + \frac{2\delta}{\gamma\mu}\right\}, \tag{24}$$

*Proof.* The results is a direct corollary of Theorems 3 and 5. $\qquad\square$

# D  ADDITIONAL EXPERIMENTS

Our code is available at `https://anonymous.4open.science/r/86315`.

## D.1  HARDWARE

We use a cluster with the following hardware: AMD EPYC 7552 48-Core CPU, 512GiB RAM, NVIDIA A100 80GB GPU, 200Gb storage space.

## D.2  EXPERIMENTAL SETUP FOR MEAN ESTIMATION PROBLEM

We consider 150 clients with data distributed as follows: the first 5 workers have data from $\mathcal{D}_1$ (the first group of clients), the next 95 workers have data from $\mathcal{D}_2$ (the second group of clients), and the remaining 50 clients have data from $\mathcal{D}_3$ (the third group of clients). Each client has 1000 samples from the corresponding distribution, and the target client has additional 1000 samples for validation, i.e., for solving the problem in Line 7. The dimension of the problem is $d = 10$. Parameters that are the same for all experiments: number of peers $= 150$, number of samples $= 1000$, batch size $= 100$, learning rate $= 0.01$, number of steps for Mirror Descent $= 50$. For `FedAvg`, the number of sampled clients $K$ is chosen from the set $\{5, 10\}$.

## D.3  RESULTS WITHOUT ADDITIONAL VALIDATION DATASET

For `MeritFed` each worker calculates stochastic gradient using a batch size of 40; then the server performs 10 steps of Mirror Descent (or its stochastic version) with a batch-size of 30 (in case of stochastic version) and a learning rate of 0.1 to update weights of aggregation, and then performs a model parameters update with a learning rate of 0.01. The plots are averaged over 3 runs with different seeds. Additionally, accuracy plots show standard deviation.

In this section, we provide experiments without an additional dataset. Instead, we use the target client's train dataset to approximately solve the problem in Line 7. The results are provided in Figures 8-11 (image classification) and Figures 12-15 (text classification). They show that `MeritFed`'s behavior with and without additional validation data is almost the same. Thus, these preliminary results give evidence that our method can be efficient in practice even when an extra validation dataset is unavailable.

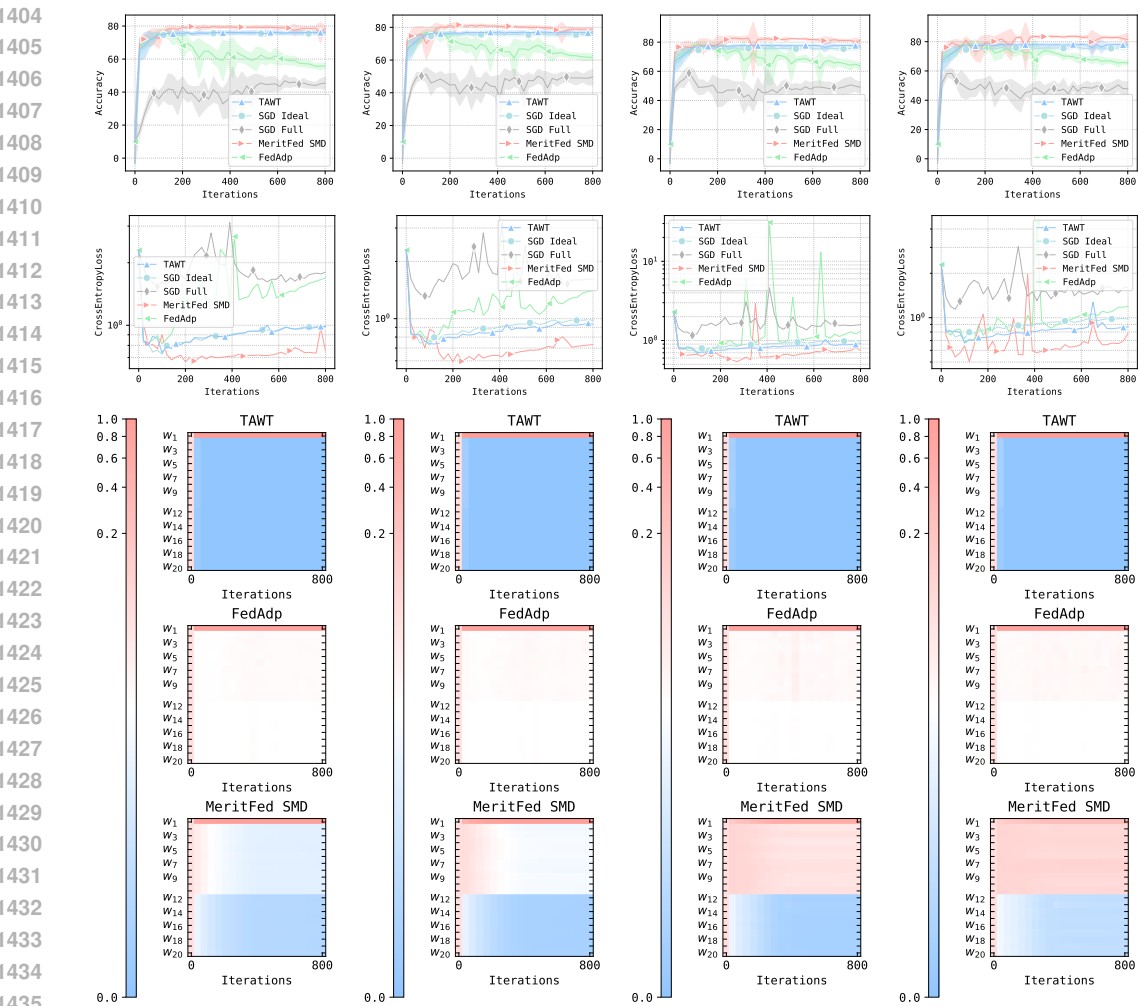

Figure 8: CIFAR10: $\alpha = 0.5$

Figure 9: CIFAR10: $\alpha = 0.7$

Figure 10: CIFAR10: $\alpha = 0.9$

Figure 11: CIFAR10: $\alpha = 0.99$

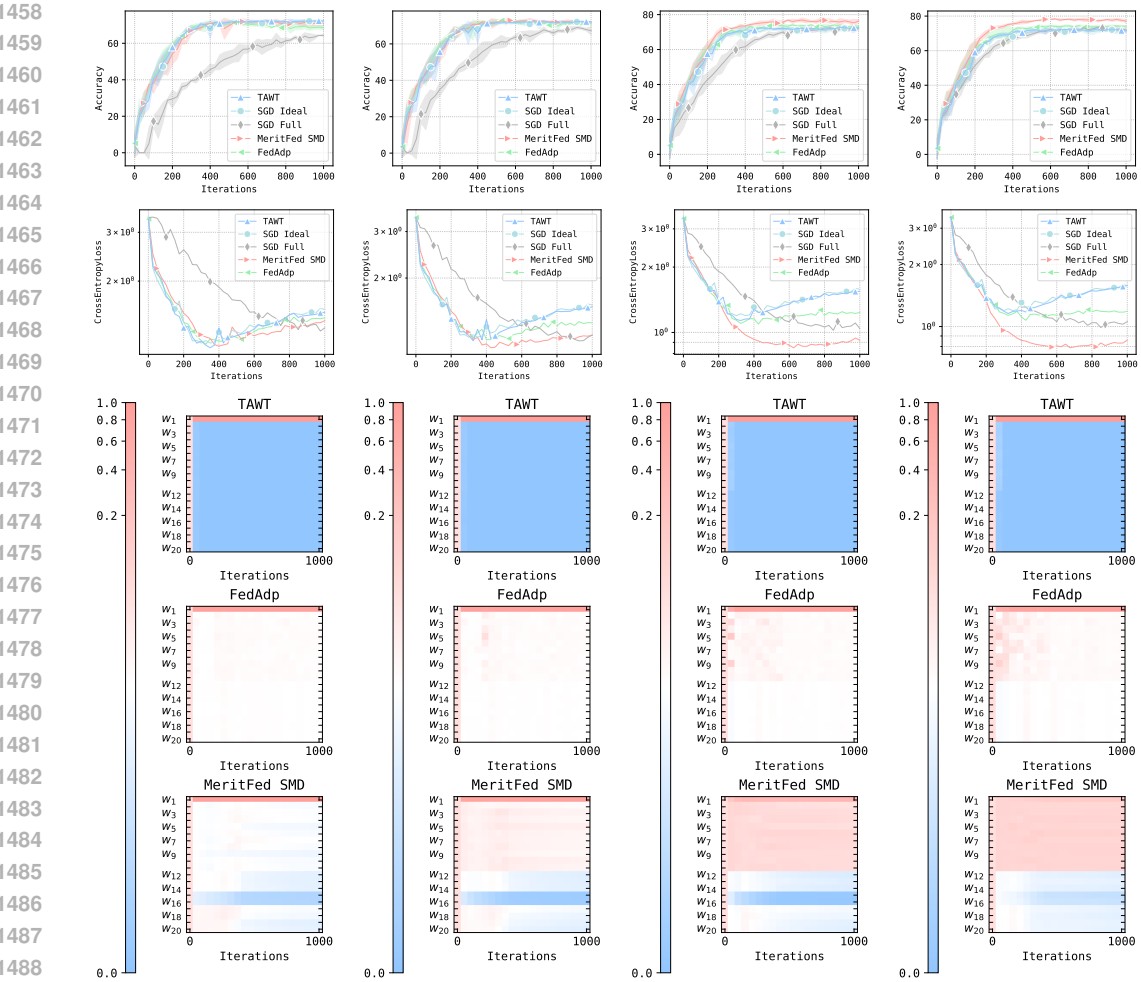

Figure 12: GoEmotions: $\alpha = 0.5$

Figure 13: GoEmotions: $\alpha = 0.7$

Figure 14: GoEmotions: $\alpha = 0.9$

Figure 15: GoEmotions: $\alpha = 0.99$

### D.4 MISSING DETAILS FOR MEDMNIST EXPERIMENTS

Complete dataset-worker mapping is OrganSMNIST, OrganAMNIST, OrganCMNIST, PathMNIST, DermaMNIST, OCTMNIST, PneumoniaMNIST, RetinaMNIST, BreastMNIST, BloodMNIST, TissueMNIST. OrganSMNIST worker is the target one.

We employ the same hyperparameters as specified in (Yang et al., 2021), including an input resolution of 28x28, ResNet-18 architecture, entropy loss, a batch size of 128, and the Adam optimizer with an initial learning rate of 0.001. This setup is run for 100 epochs, with the learning rate decreased by a factor of 0.1 after 50 and 75 epochs. Additionally, we expand the number of channels for grayscale images, as originally done by the authors.

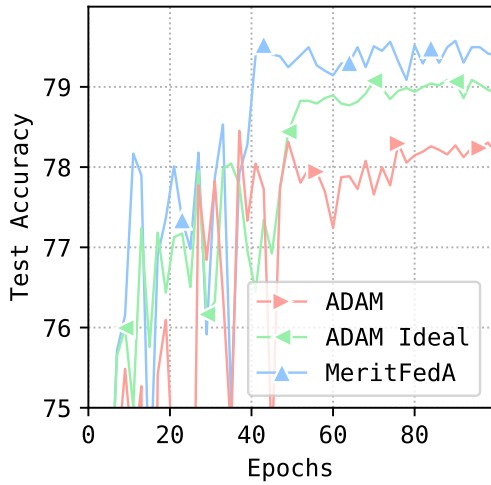
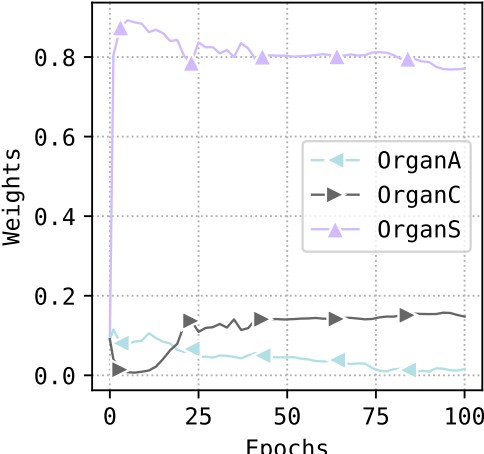

Figure 16: Test Accuracy for OrgansMNIST

Figure 17: Evolution of Relevant Weights

OrganSMNIST worker is the target one. For the `ADAM Ideal` baseline, we use only the gradients from the target client and ignore the others. Moreover, we employ the same hyperparameters as specified in (Yang et al., 2021). See For `ADAM` baseline, we aggregate gradients uniformly from the first three workers, then proceed with the Adam step. For `MeritFed`, we maintain the same parameters but adjust the learning rate schedule to reduce after 40 and 75 epochs. The mirror descent learning rate is set at 0.1, with five iterations. To enable a fair comparison, we incorporate our adaptive aggregation technique into Adam optimizer, obtaining `MeritFedA`. It adaptively aggregates gradients before performing the Adam update. The gradient with respect to the weights is obtained by deriving the Adam update formula, where the gradient is replaced with its weighted counterpart. This derived gradient is then used to update the weights of aggregation via Mirror Descent. The experimental results, depicted in Figures 16 and 17 demonstrate the superior performance of `MeritFed` and its capability to identify workers that are beneficial for training.

## D.5 ROBUSTNESS AGAINST BYZANTINE ATTACKS

`MeritFed` is robust to Byzantine attacks since our proof of Theorem 1 does not make any assumptions on the vectors received from the workers having different data distribution than the target client. This means that any worker $i \notin \mathcal{G}$ can send arbitrary vectors at each iteration, and `MeritFed` will still be able to converge. Moreover, `MeritFed` can tolerate Byzantine attacks even if Byzantine workers form a majority, e.g., the method converges even if all clients are Byzantine except for the target one.

To test the Byzantine robustness of our method on the mean estimation problem, we chose the total number of peers equal to 55 with the 50 clients being malicious. Malicious clients know the target distribution of the first 5 client and use it for performing IPM (with parameter $\varepsilon_{\text{IPM}} = 0.1$) (Xie et al., 2019) and ALIE (with parameter $z_{\text{ALIE}} = 100$) (Baruch et al., 2019) attacks. We also consider the Bit Flipping[4] (BF) and the Random Noise[5] (RN) attacks. The following choice of parameters is used: each client has 1000 samples from the corresponding distribution. The dimension of the problem is $d = 10$, learning rate $= 0.01$, number of steps for Mirror Descent $= 10$, learning rate for Mirror Descent $= 3.5$.

The results are presented in Figures 18-21. As expected, `SGD Full` does not converge under the considered attacks, and `SGD Ideal` shows the best results since, by design, it averages only with non-Byzantine workers. `FedAdp` has poor performance under ALIE attack and is quite unstable under RN attack. As in other experiments, `TAWT` is very biased towards the target client, which helps `TAWT` to tolerate Byzantine attacks, but it does not take extra advantage of averaging with clients

---

[4]Byzantine workers compute stochastic gradients $g_i^k$ and send $-g_i^k$ to the server.

[5]Byzantine workers compute stochastic gradients $g_i^k$ and send $g_i^k + \sigma \xi_i^k$ to the server, where $\xi_i^k \sim \mathcal{N}(0, \boldsymbol{I})$ and $\sigma = 1$.

having the same distribution. Finally, `MeritFed` consistently shows comparable results to `SGD Ideal`.

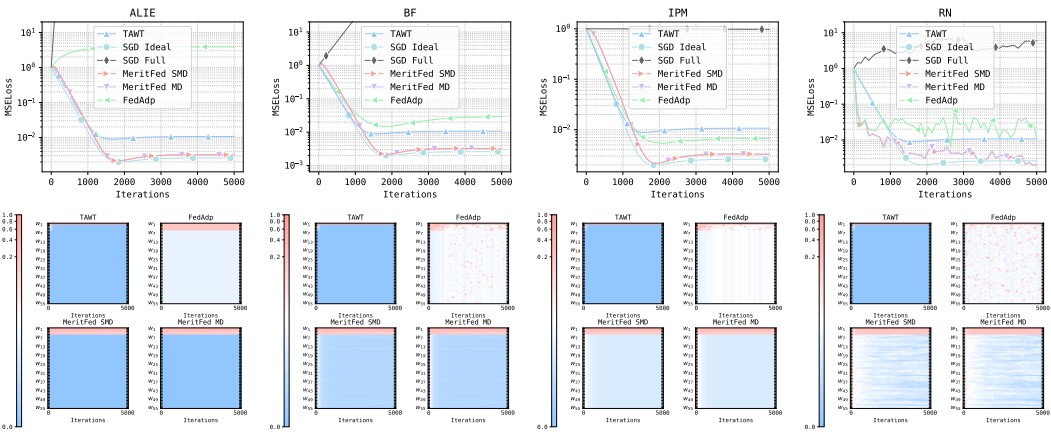

Figure 18: ALIE          Figure 19: BF          Figure 20: IPM          Figure 21: RN

### D.6   RESNET18+CIFAR10

**Image classification: CIFAR10 + ResNet18.** This part is devoted to image classification on the CIFAR10 (Krizhevsky et al., 2009) dataset using ResNet18 (He et al., 2016) model and cross-entropy loss. We consider 20 clients with data distributed as follows: the first worker has data from $\mathcal{D}_1$ (the first group of clients), the next 10 workers have data from $\mathcal{D}_2$ (the second group of clients), and the remaining 9 clients have data from $\mathcal{D}_3$ (the third group of clients). Specifically, the target client's objective is to classify the first three classes: 0, 1, and 2. This client possesses data with these three labels. The following ten workers (second group) also have datasets where a proportion, denoted by $\alpha \in (0, 1]$, consists of classes from the set $0, 1, 2$, while the remaining $1 - \alpha$ portion includes classes from the set $3, 4, 5$. The remaining clients (third group) have data from the rest, e.g., $6, 7, 8, 9$ labeled. The data is randomly distributed among clients without overlaps, adhering to the aforementioned label restrictions. For `MeritFed` each worker calculates stochastic gradient using a batch size of 75; then the server performs 10 steps of Mirror Descent (or its stochastic version) with a batch-size of 90 (in case of stochastic version) and a learning rate of $0.1$ to update weights of aggregation, and then performs a model parameters update with a learning rate of $0.01$. We normalize images (similarly to (Horváth & Richtárik, 2020)). Since an additional validation dataset can be used by `MeritFed`, we cut 300 samples of each target class $(0, 1, 2)$ off from the test data. Accuracy and loss are calculated on the rest of the test data, including labels 0, 1, and 2, modeling the case when the target client aims to classify samples with these labels.

The results are provided in Figures 22-25, where we show how accuracy and cross-entropy loss change for different methods and different values of $\alpha$, which measures the similarity between data distributions of the target client and the second group of clients, and the evolution of the aggregation weights. In all settings, `MeritFed` outperforms `SGD Ideal` and other baselines regardless of $\alpha$. In all cases, the weights are almost the same for all workers during the few initial steps (even if workers have quite different distributions like for the last nine clients). This phenomenon can be explained as follows: if we have two different convex functions with different optima (e.g., two quadratic functions), then for a far enough starting point, the gradients of those functions will point roughly in the same direction. Therefore, during a few initial steps, both gradients are useful and the method gives noticeable weights to both. However, once the method comes closer to the optima, the gradients become noticeably different, and after a certain stage, the gradient of the second function no longer points closely towards the optimum of the first function. Therefore, starting from this stage, `MeritFed` assigns a smaller weight to the gradient of the second function. Going back to Figures 22-25, we see a similar behavior: for $\alpha = 0.5$, the advantages of collaboration with clients 2-11 disappear after a certain stage since the method reaches the region where two distributions become noticeably different. In contrast, when $\alpha = 0.99$, those workers have a very close distribution

to the target worker, and therefore, their stochastic gradients remain useful during the whole learning process. `FedAdp` is biased to the target client and assigns almost identical weights to either clients with similar or dissimilar distributions, which results in an accuracy decrease at the end of the training, in contrast to `MeritFed`, which tracks and maintains less weights to non-beneficial clients. `TAWT` is much more biased to the target client, which makes it almost identical to `SGD Ideal`.

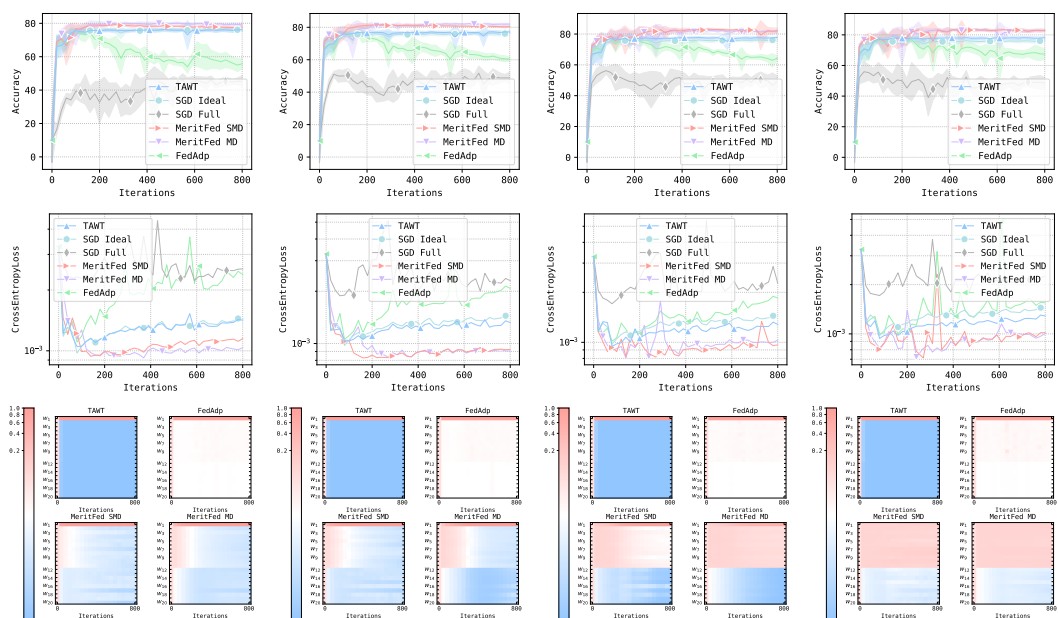

Figure 22: CIFAR10 (extra val.): $\alpha = 0.5$

Figure 23: CIFAR10 (extra val.): $\alpha = 0.7$

Figure 24: CIFAR10 (extra val.): $\alpha = 0.9$

Figure 25: CIFAR10 (extra val.): $\alpha = 0.99$.

### D.7 RESNET18+CIFAR10: 40 WORKERS

In the mean estimation problem, we generate the data and can control the number of workers. Therefore, for this problem we have many clients participating in the training.

However, for the other two tasks, datasets are fixed. Therefore, we limited the number of workers to 20 to have enough data on each client (given the splitting strategy) without repetition. That is, each data sample (image or tokens) from the original datasets belongs to no more than 1 client. Therefore, to run experiments with more workers we either need to have more data or allow repetitions in data on the clients.

In the additional experiments, we have 40 clients where the new 20 clients are just copies of the first 20 clients. The experimental setup follows the same data partitioning idea as presented in the paper and deals with four values of heterogeneity values across clients $\alpha$. For `MeritFed` each worker calculates stochastic gradient using a batch size of 75; then the server uses Mirror Descent (or its stochastic version) with a batch-size of 90 (in case of stochastic version) and a learning rate of 0.1 to update weights of aggregation, and then performs a model parameters update with a learning rate of 0.01.

The results presented on Figures 26-29. Overall, the conclusions are consistent with what we have in the experiment with 20 workers, further supporting the scalability of `MeritFed`.

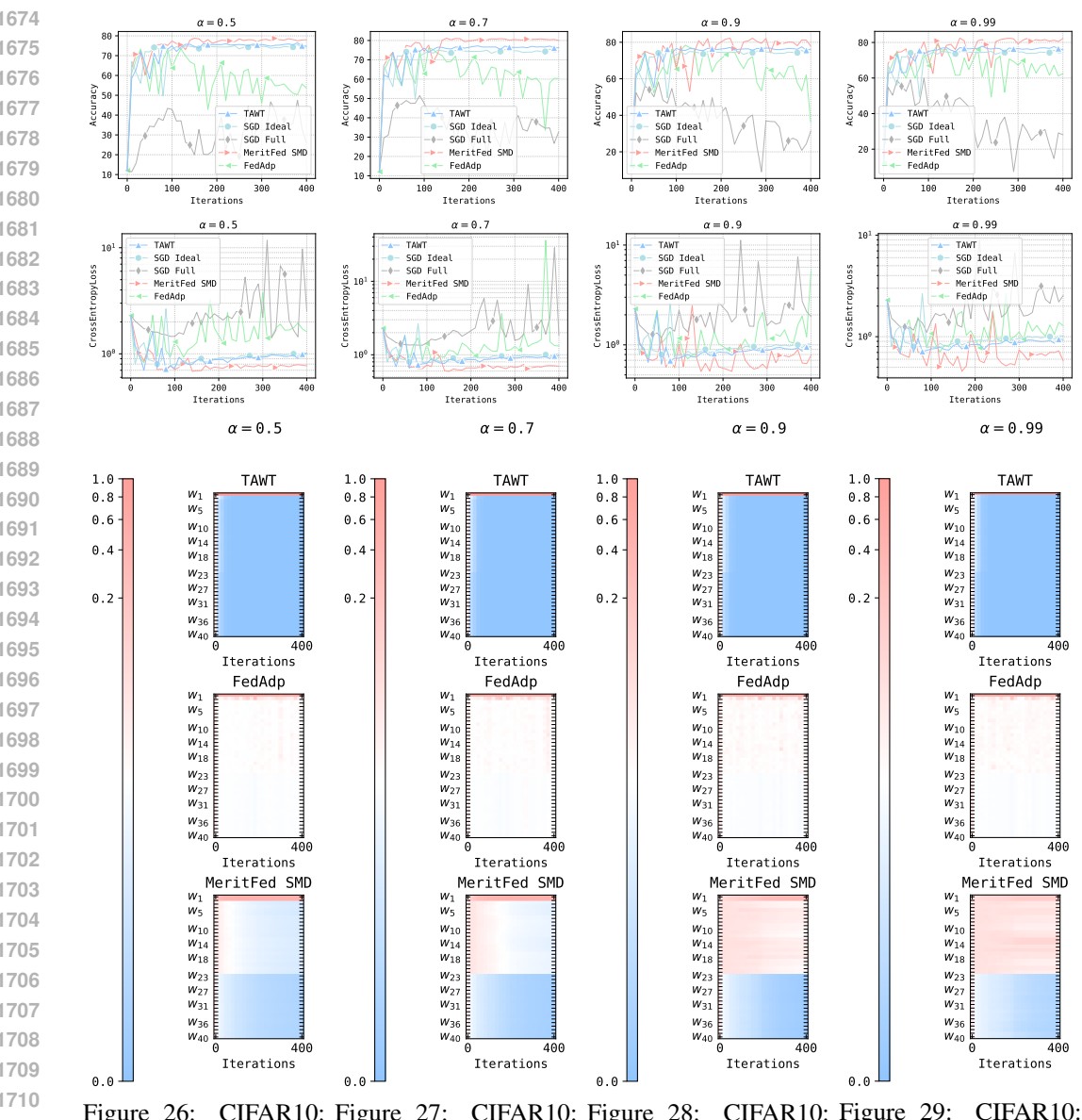

Figure 26: CIFAR10: $\alpha = 0.5$

Figure 27: CIFAR10: $\alpha = 0.7$

Figure 28: CIFAR10: $\alpha = 0.9$

Figure 29: CIFAR10: $\alpha = 0.99$.

