# OpenReview forum: "Selective Collaboration for Robust Federated Learning"
_ICLR.cc/2026/Conference — ICLR 2026 Conference Withdrawn Submission_

### Official Review · Reviewer_m68h · 2025-10-19

**Soundness:** 2
**Presentation:** 2
**Contribution:** 2
**Rating:** 2
**Confidence:** 3

**Summary:**

The paper proposes MeritFed to address data heterogeneity in FL. It dynamically adjusts aggregation weights based on each client’s contribution relevance, formulated through an auxiliary optimization problem solved approximately via zeroth-order mirror descent. This method ensures convergence under mild assumptions and shows robustness to adversarial or outlier clients. Experiments on mean estimation, GoEmotions, and MedMNIST show that MeritFed effectively identifies beneficial collaborators and outperforms baselines, achieving better performance and rebostness to Byzantine attacks.

**Strengths:**

1. The presentations of this paper, including figures and tables, are good.
2. This paper proposes to address a subproblem by using zeroth-order mirror descent or its accelerated version, which enhances clients’ privacy protection.
3. It provides theoretical analysis and empirically show the advantages of their proposed MeritFed.

**Weaknesses:**

W1: The goal of this work is not enough convincing due to the concerns about fairness and practicality. (See Q1 for specfic questions).

W2: In Section 1.3 Related Work, the authors introduced many existing works about reducing communication cost. However, they did not show or discuss anything about communication cost for their proposed MeritFed except for future work (correct me if I am wrong). This kind of inconsistency negatively affects the writing of this paper.

W3: The theory analysis could be improved further, especially about $\delta$.

W4: In the main paper, the description and discussion of experiment results is not enough and even not provided in some places, which results in unclear performance of the proposed MeritFed.

**Questions:**

Q1: For general or personalized FL frameworks, each participating client benefits directly from the trained model. However, the goal of this paper is fundamentally different. In the proposed setting, a group of clients collaboratively assist a target client in training its model, which offers no direct benefit to the helper clients. Can the authors clarify in what real-world scenarios such a one-sided collaboration would be practical or realistic? Moreover, this setup raises fairness and participation concerns: clients with limited computational or communication resources may be required to help others frequently but gain nothing in return.

Q2: MeritFed requires the server to store $n$ vectors at each iteration, which is a memory complexity of $O(n*d)$, where $d$ is model dimension. Can the authors provide detailed results or analysis about the memory usage in your experiments to prove that the memory cost is not an issue for modern servers? Moreover, if $d$ is quite large, such as million-level or billion level large language models, what will be the memory cost then?

Q3: I did not find any possible extension to partial client participation, which may limit the practicality of MeritFed in real-word applications. If there are some unavailable clients in your FL system, how the proposed MeritFed will be affected by that? Does the MeritFed still work?

Q4: In Line 336, $\delta$ is defined as the accuracy of solving the problem in Line 9 of Algorithm 1. Based on the problem related to zeroth-order optimization, $\delta$ should be related to model dimension $d$. Hence, can the authors discuss $\delta$ and explain how to get a small $\delta$?

Q5: Are the meaning of $\alpha$ in equation (8) and Figure 4 the same?

See weaknesses above.

---

### Official Review · Reviewer_JJCz · 2025-10-20

**Soundness:** 2
**Presentation:** 3
**Contribution:** 2
**Rating:** 6
**Confidence:** 4

**Summary:**

This paper aims to solve the collaboration problem in federated learning (FL). The authors introduce a dynmamic weight assigning method for FL based on the clients' data distribution. They establish the convergence guarantee and emplically validate the effectiveness of the proposed method.

**Strengths:**

1. This work define the weight selection in FL as an optimization problem, which is  more principled than common heuristic methods.

2. This work provides convergence guarantees for the design under commonly used assumptions. The results prove the same or better convergence speed than an oracle method.

3. This work provides sufficient empirical experiments across several tasks (vision, NLP, medical), showing the robustness and improvements of the proposed method.

**Weaknesses:**

The main concern lies in the solution-solving part of MeritFed and the evaluation metric.

1. MeritFed requires iteratively solving an auxiliary problem via a zeroth-order oracle, which introduces additional communication and computation overhead in each global iteration. This may cause issues since communication and computation are often the bottleneck of clients.

2. MeritFed relies on a dataset at the target client to solve the auxiliary problem, which can be problematic. Using the training data for this purpose risks overfitting, while assuming a validation set is a strong practical limitation. This makes the method less practical in real-world applications.

3. The experiments use "iterations" as the primary metric for comparison, but MeritFed may require more time for each iteration, making the comparison unfair in some cases. A further discussion with the wall-clock time is required.

**Questions:**

1. How does MeritFed compare with baselines with the wall-clock time as the x-axis?

2. Why not compare MeritFed to a personalized version of the mentioned works, such as FedProx, regularizing the model updating with the local model weight?

3. It seems to me that MeritFed implicitly forms a "soft cluster" of beneficial clients around the target. Why was there no comparison to explicit Clustered FL algorithms?

---

### Official Review · Reviewer_HdXN · 2025-10-29

**Soundness:** 2
**Presentation:** 2
**Contribution:** 2
**Rating:** 2
**Confidence:** 3

**Summary:**

This paper addresses the challenge of heterogeneous and potentially adversarial client data in federated learning. Instead of uniform aggregation (as in FedAvg), the authors propose MeritFed, a method that dynamically learns aggregation weights for each client at every communication round based o merit. The key idea is to approximate the solution of an auxiliary optimization problem where the aggregation weights are chosen to minimize the target client's expected loss.

**Strengths:**

The problem and motivation are clear. The paper addresses a well-known issue: not all clients are helpful in heterogeneous FL, and naive aggregation can degrade target performance. The focus on beneficial collaboration is timely and grounded.

**Weaknesses:**

1. The main contribution for the proposed method is merit-based weight. However, it is not easy to calcualte the weights. 1) We do not have public dataset in the server, which is expected to have the same distribution as target client's data. 2) It has extra computation and memory requirements. 3) It could introduce other issues such as fairness when this public dataset is biased.

2. I wonder what is the final convergence rate and its comparison with baselines theoretically.

**Questions:**

1. It is good to have a dynamic weight, but I have concerns about the fairness for participated clients for 1) the quality of the public data and 2) the re-calculation process might introduce bias?

---

### Official Review · Reviewer_nYcw · 2025-11-01

**Soundness:** 3
**Presentation:** 2
**Contribution:** 1
**Rating:** 2
**Confidence:** 4

**Summary:**

This work developed an algorithm, MeritFed, that aggregates information from multiple clients to help learning a model for one specific client. The server distributes the model to all clients, collects their gradients, and finds the combining weights for the gradients so that the combined gradient can best suit one particular client’s objective. Theoretical convergence analysis is provided. Experiments conducted on synthetic as well as real-world data show that MeritFed can perform better than some other baselines.

**Strengths:**

1. The paper shows the convergence (Theorems 1 and 2) and benefits (Theorem 2) when there exist similar clients (similar in the sense of Assumption 2).

2. Experiments are conducted for various problems to show the advantages of the proposed method.

**Weaknesses:**

1. The problem setting is strange. It remains unclear why a server would be dedicated to serving one specific client using information from many other clients. In federated learning, the hope is to learn good model(s) so all clients can benefit to some extent. If each client has some specific needs due to data heterogeneity, then it would make more sense to conduct personalized federated learning.

- Line 9 of Algo.1 is on the server side and L247 mentioned that the server uses a duplicate of Client 1’s data. This bluntly violates the principle in FL of not leaking data outside of Client 1. The experiment uses additional validation data, which may not exist in real-world applications.

2. The writing can be improved. There are several notation issues and questionable statements in the paper. For notations:

- Eq.(5) It is unclear why $\xi$ needs to be hold (c.f., Eq.(1)).
- Where does the $n$ come from for the numerator in L265? And where did $R$ appear before L284?
- $f^*$ in L323, $\sigma$ in L331 (&339) undefined.

For statements

- L169: It is unclear why they are equivalent. They are very different problems.
- L401: How come MeritFed and FedProx have no local steps? Isn’t Line 6-7 in Algo. 1 local updates?

3. The experiments lack proper comparison. The baselines are very restricted. MeritFed is highly related to personalized FL methods in the literature and imposing constraints (L377) for the choices of compared baselines is unnecessarily limited. It would make sense to compare to, for example, Per-FedAvg (Fallah et al., 2020), Ditto (Li et al., 2021), PFedMe (T Dinh et al., 2020), FedFomo (Zhang et al., 2020), FedALA (Zhang et al., 2023), FedeRiCo (Sui et al., 2022).

Ref:

- Li, T., Hu, S., Beirami, A. and Smith, V., 2021, July. Ditto: Fair and robust federated learning through personalization. In *International conference on machine learning* (pp. 6357-6368). PMLR.
- T Dinh, C., Tran, N. and Nguyen, J., 2020. Personalized federated learning with moreau envelopes. *Advances in neural information processing systems*, *33*, pp.21394-21405.
- Sui, Y., Wen, J., Lau, Y., Ross, B.L. and Cresswell, J.C., 2022. Find your friends: Personalized federated learning with the right collaborators. *arXiv preprint arXiv:2210.06597*.
- Zhang, J., Hua, Y., Wang, H., Song, T., Xue, Z., Ma, R. and Guan, H., 2023, June. Fedala: Adaptive local aggregation for personalized federated learning. In *Proceedings of the AAAI conference on artificial intelligence* (Vol. 37, No. 9, pp. 11237-11244).

**Questions:**

Q1: Why would we want to serve one client and one client only? Why not conduct personalized FL instead?

Q2: What is the performance of the baselines mentioned above?

---

### Author Response · Authors · 2025-11-18
**Request for quality check**

Dear Chairs,

After carefully reviewing the comments, we believe that many of the reviewers’ concerns are already addressed in the paper. Unfortunately, the reviews do not accurately reflect the content or contributions of our work, and we find that the overall quality of the feedback is below the standard we expected. To make this concrete, we list representative corrections below; line numbers refer to the main text.

## Reviewer nYcw

>**The problem setting is strange.**
Line 051-056, 067-078 discuss the setting and provide real world scenarios

>**Line 9 of Algo.1 is on the server side and L247 mentioned that the server uses a duplicate of Client 1’s data.**
It contradicts the paper content: Line 247 says nothing about the server.
Lines 251-254 explicitly state that the target client dataset should be stored only on
the target client and provide a solution via ZO MD.

>**3) The experiments lack proper comparison.**
Line 376-377 explains on selected baselines.

## Reviewer HdXN

>**The main contribution for the proposed method is merit-based weight. However, it is not easy to calcualte the weights.**
1) >**We do not have public dataset in the server, which is expected to have the same distribution as target client's data.**
Lines 251-254 explicitly state that the target client dataset should be stored only on the target client and provide a solution via ZO MD. Lines 246-248 addresses the distribution concern.
2) >**It has extra computation and memory requirements.**
Line 286-290 addresses the concern.
3) >**It could introduce other issues such as fairness when this public dataset is biased.**
Lines 251-254 explicitly state that the target client dataset should be stored only on the target client and provide a solution via ZO MD. Moreover, in Lines 063-066 we allow adversarial participants, so we can provably benefit even them.**


>**I wonder what is the final convergence rate and its comparison with baselines theoretically.**
Within the standard oracle models, our rates match the known lower bounds for the corresponding method classes, so no theoretical baseline can be strictly faster up to constants unless a different oracle model is assumed. The paper states the rates in Theorems 1 and 2.

## Reviewer m68h

>**W1, Q1:**
Line 051-056, 067-078 discuss the setting and provide real world scenarios. While the helper clients in our setting do not obtain a direct model benefit, such one-sided collaborations are already common when data (or signals derived from data) are exchanged for compensation. At the consumer level, users opt in to share browsing or device-usage data in return for rewards (e.g., Brave Rewards/BAT, Nielsen Computer & Mobile Panel, Gener8, Tapestri) or via “data unions” such as Swash that collectively monetize streams of user data. At the enterprise level, data-licensing is a mature industry: registered data brokers aggregate and sell datasets (e.g., Experian Marketing Services; see also the FTC’s report on data brokers and state broker registries). In this light, our FL setting is practical whenever a “target” party values a tailored model and compensates helpers (e.g., micropayments, credits, or access to aggregate analytics) without exposing raw data. This addresses fairness and participation: helpers are paid according to participation policies (e.g., per-round budgets, opt-out, and contribution-aware rewards), while the protocol preserves privacy via federated updates. See also Line 076-068 on privacy.

>**W2:**
That is not true. Lines 103-111 discuss communication bottlenecks, and clearly state that our technique is orthogonal. Lines 181-192 discuss concurrent papers. Line 483  leaves communication efficiency for future works.**

>**W3:**
The statement is unsupported and it does not specify what is missing or incorrect. In our paper $\delta$ is the inner-solve accuracy parameter; the theorems already state how the bounds depend on $\delta$. Absent a concrete target (which bound to tighten, which assumption to relax), the point is not actionable.

>**W4:**
The claim that discussions of experimental results are “even not provided” is not true. The manuscript already contains text passages that interpret the figures (main: Lines 417-425, 445-448; appendix: 1402-1403, 1558-1568, 1606-1626).

Given these circumstances, we respectfully request an Area Chair quality check of the reviews before we enter the author–reviewer discussion. Specifically, we ask that the reviewers either revise their comments with precise citations and actionable feedback. If a quality check is not possible, please let us know; in that case we would prefer to withdraw rather than participate in a discussion that cannot be made evidence based.

Thank you for your time and for managing the review process.

Best regards,

Authors of Submission # 10809

---

### Note · Authors · 2025-12-11

I have read and agree with the venue's withdrawal policy on behalf of myself and my co-authors.